# Hospital services utilisation and cost before and after COVID-19 hospital treatment: Evidence from Indonesia

**Muhammad Fikru Rizal[1,2], Firdaus Hafidz●[1]\*, Gilbert Renardi Kusila●[1], Wan Aisyiah[3], Dedy Revelino[3], Erzan Dhanalvin[3], Ayunda Oktavia[3], Ilyasa[3], Citra Jaya[3], Benjamin Saut[3], Mahlil Ruby[3]**

**1** Department of Health Policy and Management, Faculty of Medicine, Public Health, and Nursing, Universitas Gadjah Mada, Yogyakarta, Indonesia, **2** Centre for Health Economics, Monash Business School, Monash University, Melbourne, Victoria, Australia, **3** Badan Penyelenggara Jaminan Sosial Kesehatan, Jakarta, Indonesia

\* hafidz.firdaus@ugm.ac.id

**Data Availability Statement:** All relevant data used to produce the graphs in the paper are provided in S1 and S2 Tables. The raw data that support the findings of this study are available from Indonesia's

## Abstract

### Objective

To estimate hospital services utilisation and cost among the Indonesian population enrolled in the National Health Insurance (NHI) program before and after COVID-19 hospital treatment.

### Methods

28,159 Indonesian NHI enrolees treated with laboratory-confirmed COVID-19 in hospitals between May and August 2020 were compared to 8,995 individuals never diagnosed with COVID-19 in 2020. A difference-in-difference approach is used to contrast the monthly all-cause utilisation rate and total claims of hospital services between these two groups. A period of nine months before and three to six months after hospital treatment were included in the analysis.

### Results

A substantial short-term increase in hospital services utilisation and cost before and after COVID-19 treatment was observed. Using the fifth month before treatment as the reference period, we observed an increased outpatient visits rate in 1–3 calendar months before and up to 2–4 months after treatment (p<0.001) among the COVID-19 group compared to the comparison group. We also found a higher admissions rate in 1–2 months before and one month after treatment (p<0.001). Consequently, increased hospital costs were observed in 1–3 calendar months before and 1–4 calendar months after the treatment (p<0.001). The elevated hospital resource utilisation was more prominent among individuals older than 40. Overall, no substantial increase in hospital outpatient visits, admissions, and costs beyond four months after and five months before COVID-19 treatment.

Social Security Administrative Body of Health (BPJS-Kesehatan), which were used under license for the current study, and so are not publicly available. Researchers can request access to the data by contacting BPJS-Kesehatan at ppid@bpjs-kesehatan.go.id. Access to the dataset is subject to approval by BPJS-Kesehatan

**Funding:** This study is fully funded by Indonesia's Social Security Administrative Body of Health (BPJS-Kesehatan). Award/grant number 456/BA/0621. The funders had no role in study design, data collection and analysis, decision to publish, or preparation of the manuscript.

**Competing interests:** The authors have declared that no competing interests exist.

## Conclusion

Individuals with COVID-19 who required hospital treatment had considerably higher health-care resource utilisation in the short-term, before and after the treatment. These findings indicated that the total cost of treating COVID-19 patients might include the pre- and post-acute period.

## Introduction

As the COVID-19 pandemic continues to become a global health issue, the understanding of its consequences beyond the respiratory system and the acute treatment for the disease itself has become a major concern. Several studies found that COVID-19 infection does not only affect the respiratory system but also haematological, cardiovascular, neurological, and gastro-intestinal organs [1]. It has also been found that older people and those with chronic illnesses are more vulnerable to developing severe and critical COVID-19 infections. Moreover, these groups also have a higher mortality rate [2]. A study conducted in the United States found that COVID-19 hospitalisation was six times higher in patients with pre-existing comorbidities than in those without [3]. In Indonesia, about 12% of the hospitalised patients died in the first three months of the pandemic, with higher rate among older people, male, and those with multiple comorbidities [4]. A more recent study indicated that districts with higher proportion of elderly and lower healthcare capacity had higher rate of COVID-19 mortality [5].

Apart from our increasing knowledge related to the risk factors for COVID-19 and its mortality, our understanding of the clinical sequelae of COVID-19 infections has grown over time. Due to its multi-organ involvement, some COVID-19 symptoms can persist for up to six months after infection remission or more, which is defined as "long COVID" or "post-COVID" syndrome [6]. A meta-analysis conducted by Chen et al. found that fatigue, memory problem, and respiratory issue are the three most prevalent symptoms [7]. Current evidence indicates that older patients, those with more than five symptoms and hospitalised, and the Asian population are also identified to be more likely to experience long covid [6–8].

The daily activities and quality of life deteriorate among the people who suffer from long covid, with physical limitations and emotional stress included [9, 10]. The post-covid examination and therapies are then needed in a large proportion of recovered patients. Furthermore, routine pulmonary rehabilitation and some medication consumptions were marked effective in decreasing dyspnoea among long covid patients [11]. From a policymaker's perspective, the importance of long-covid cannot be understated. In the short run, the health system in each country needs to anticipate the potential change in healthcare needs through better allocation of resources and investment [12]. However, evidence on the potential impact of COVID-19 infections on subsequent healthcare use and costs is scarce, especially in low-resource settings.

Findings from National Health Services (NHS) in England indicate that COVID-19 hospitalisation increases the readmission rate and subsequent diagnosis of respiratory diseases, diabetes, and cardiovascular diseases [13]. For mild infections, a study in Norway found a short-term elevation of primary care use, mostly for a respiratory and general or unspecified condition, but no effect on specialist care. This change, however, mostly reverted after three months [14, 15]. While these studies used a large administrative database, hence minimising reporting bias, none of them reported the costs associated with pre- or post-COVID changes in healthcare use.

This paper aimed to fill the literature gap by estimating the change in utilisation and cost of hospital services before and after individuals were treated in hospitals with laboratory-confirmed COVID-19. We leveraged administrative National Health Insurance (NHI) data from Indonesia, a lower-middle-income country (LMIC) with about 260 million population. The Indonesian NHI is the biggest single-payer social health insurance program in the world, covering more than 90% of the Indonesian population [16]. Our study aims to examine the impact of COVID-19 by analysing both the direct effects on patients treated for the disease and the associated changes in hospital service utilization and costs. This dual perspective provides insights into the short- to medium-term adjustments within hospital services pre- and post-acute COVID-19 treatment periods, especially critical for understanding the pandemic's ramifications in resource-constrained settings.

## Methods

### Data source

We utilised all COVID-19-related hospital claims which were recorded in the deidentified National Health Insurance (NHI) database from January 2020 to February 2021. Patients who were in this COVID-19 claims data were linked with their regular hospital claims records dated back to December 2016. The merged dataset contains information related to patients' basic demographic characteristics and healthcare utilisation records. The basic demographic characteristics include the year of birth, sex, province where individuals were treated with COVID-19 and type of NHI membership (subsidised or non-subsidised scheme).

The utilisation records contain the month of encounter, type of services (inpatient or outpatient), primary and secondary diagnoses as comorbidity, diagnoses-related group codes (used as the basis for hospital payment), severity level, levels of hospitals (a basic type D hospital to an advanced type A hospital), amount of claims, and the discharge status (in-hospital death, routine, or against medical advice), and COVID-19 diagnosis status (suspect, probable, or laboratory-confirmed).

### Ethical clearance

Our study did not obtain the written consent from the participants, as it used an NHI database as it source. All data were fully anonymized before being accessed. Ethical clearance was obtained from the Ethical Committee at Faculty of Medicine, Public Health, and Nursing, Universitas Gadjah Mada (No KE/FK/0945/EC/2021).

### Study population

Individuals who were diagnosed based on positive RT-PCR results and treated, either in outpatient care or admitted to inpatient care, from May to August 2020 were included in the *COVID-19 group*. Individuals who were under monitoring and recorded as suspect or probable cases of COVID-19 in the previous months before laboratory confirmation were excluded. Similarly, those who died during COVID-19 hospitalisation or in subsequent months were also excluded from the analysis. This decision was made to focus on COVID-19 survivors, hence avoiding potential issue of reduced healthcare use of the COVID-19 group due to their higher post-hospital treatment mortality. For context, the recorded 6-month mortality rate of those treated with COVID-19 between May and August 2020 was 12.1%. The comparison group comprised individuals treated for COVID-19 in hospitals in February 2021 (not hospitalised in 2020), under the assumption they had not contracted the virus in 2020. We argue this is a suitable comparator for our analysis as these groups are similar (i.e. attitude towards

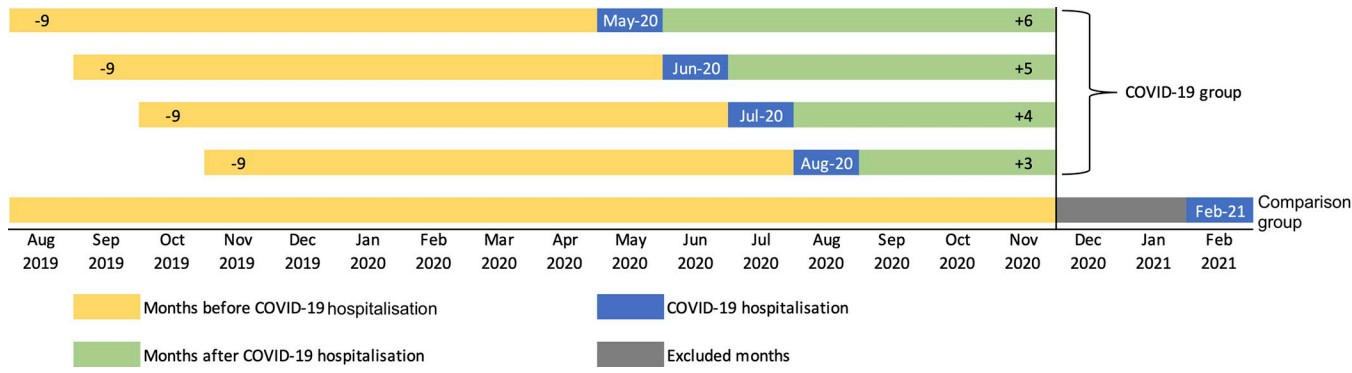

**Fig 1. Observation period for COVID-19 and comparison group.**

risk, preference, etc.). As we observed their health care utilisation in the entirety of our estimation sample from August 2019 to November 2020, the event study approach allows us to compare the outcomes between treatment and comparison groups over time. This assumption is plausible given reinfection rate within one year follow up was only about 5% [17]. Note that in the NHI database, only results from RT-PCR (Reverse Transcriptase Polymerase Chain Reaction) done by Ministry of Health-accredited laboratories are used [18].

## Study period

We considered a period of August 2019 to November 2020 in our analysis. During this period, the comparison group, those who were treated with COVID-19 in February 2021, was assumed to not yet contracted the disease. December 2020 and January 2021 were excluded due to the possibility of elevated healthcare utilisation two months before individuals in the comparison group were hospitalised with COVID-19 in February 2021. Hence, we had a maximum of six months (June 2020 to November 2020) and a minimum of three months (September to November 2020) post-COVID-19 follow-up. For pre-COVID period, we included nine months before COVID-19 hospital treatment. The grouping strategy and the period of observation used in our analyses is shown in Fig 1.

## Outcome variables

We measured the total number of hospital service utilisation, both inpatient and outpatient, and the cost of direct healthcare-related services as total reimbursement from national health insurance at the hospital (in Indonesian Rupiahs) separately for COVID-19 diagnosis and other diagnoses. Our estimates were presented in thousand Indonesian Rupiah (US$1 equals to Rp14,500). All these measures were recorded for each individual on a monthly basis. During the month of COVID-19 diagnosis and treatment, all the patients would have at least one hospital encounter.

## Statistical analysis

Baseline characteristics of the COVID-19 group (overall and separately by the month of treatment) and the comparison group were described as percentages (for categorical variables) or mean and standard deviation (for continuous variable).

Next, we employed a difference-in-difference (DiD) event-study approach as implemented by Skyrud et al. and Magnusson et al. to estimate the association between COVID-19 hospital treatment and the utilisation and costs of hospital services, before and after the treatment

[14, 15]. This technique estimated the association between an exposure and an outcome by contrasting the change in outcome among the exposed group from a given time before exposure (the reference period) to another time point with the change in outcomes among the comparison group within the same time frame. To identify causal effect of an exposure, it is assumed that in the absence of exposure, there should be no difference in outcomes trends over time between the exposed (COVID-19) group and the unexposed (comparison) group [19, 20].

Empirically, we estimated the following regression model (Eq 1).

$$Y_{it} = \Sigma_{t=Min}^{t=Max} \beta_{1t} Month_t \times Covid_i + \beta_2 Covid_i + \beta_3 Month_t + \boldsymbol{\beta_4} \boldsymbol{X_i} Month_t + \boldsymbol{\beta_5} \boldsymbol{X_i} + \varepsilon_{it} \quad (1)$$

The $Y_{it}$ variable represents the outcomes of individual i in month $t$. $Month_t$ is a categorical variable indicating the month or period of observation, with month *zero* indicating the month where individual was treated for laboratory-confirmed COVID-19 infection. *Min* and *Max* indicate how many months (or groups of months) before and after the month of hospitalisation, respectively (see Fig 1 for how we constructed these observations period). For hospital service utilisation regressions, we used the fifth month leading to the hospitalisation as the reference period. For example, May 2020 group, this reference period was December 2019 and labelled as t-5. The t0 was May 2020, the t+1 to t+6 was June to November 2020, the t-4 to t-1 was January 2020 to April 2020. For each subsequent month, the reference and comparative periods shifted accordingly".

As for cost regression, we used the period of 6 to 4 months before hospitalisation as the reference period and lumped together 9 to 7 months before the hospitalisation as pre-reference period. $Covid_i$ is a binary variable indicating whether individuals are in the COVID-19 or in the comparison group. As previously discussed, nine months period before exposure and three to six months period after exposure were used in the main analyses. Coefficient $\beta_{1t}$ capture the month specific DiD estimates measuring the association between COVID-19 hospital treatment and each of the outcomes.

Although some of our outcomes were count variables, linear modelling can still be an appropriate choice [21], hence we estimate Eq 1 using Ordinary Least Square (OLS). Next, to account for differences in characteristics between COVID-19 and comparison group, we also include set of control variables ($\boldsymbol{X_i}$) which include age, sex, severity, comorbidities diagnosed before January 2020, NHI membership segment, and province. These variables are interacted with month indicator to allow for their changing influence on outcomes over time. Severity levels in our study are defined based on the Indonesian Case-Based Groups (INA-CBGs) system, a case-mix payment system that utilizes a software grouper application. The severity level is influenced by complications and comorbidities, which are indicative of the resource intensity level required for treatment during the first treatment episode where patients were laboratory-confirmed. Heterogeneity analysis was conducted by splitting the sample into younger (<40 years old) and older (≥40) population, as well as by gender. The age cut-off was considered given the risk of major comorbidities such as cardiovascular diseases and cancer known to increase exponentially after the age of 40 [22].

## Results

### Study participants

Of 247,650 NHI members with COVID-19-related hospital encounters between January 2020 and February 2021, 146,240 were confirmed to be infected by COVID-19 based on their RT-PCR results, while the others were categorised as suspect or probable cases only. We excluded those who died during or after COVID-19 treatment and those who were

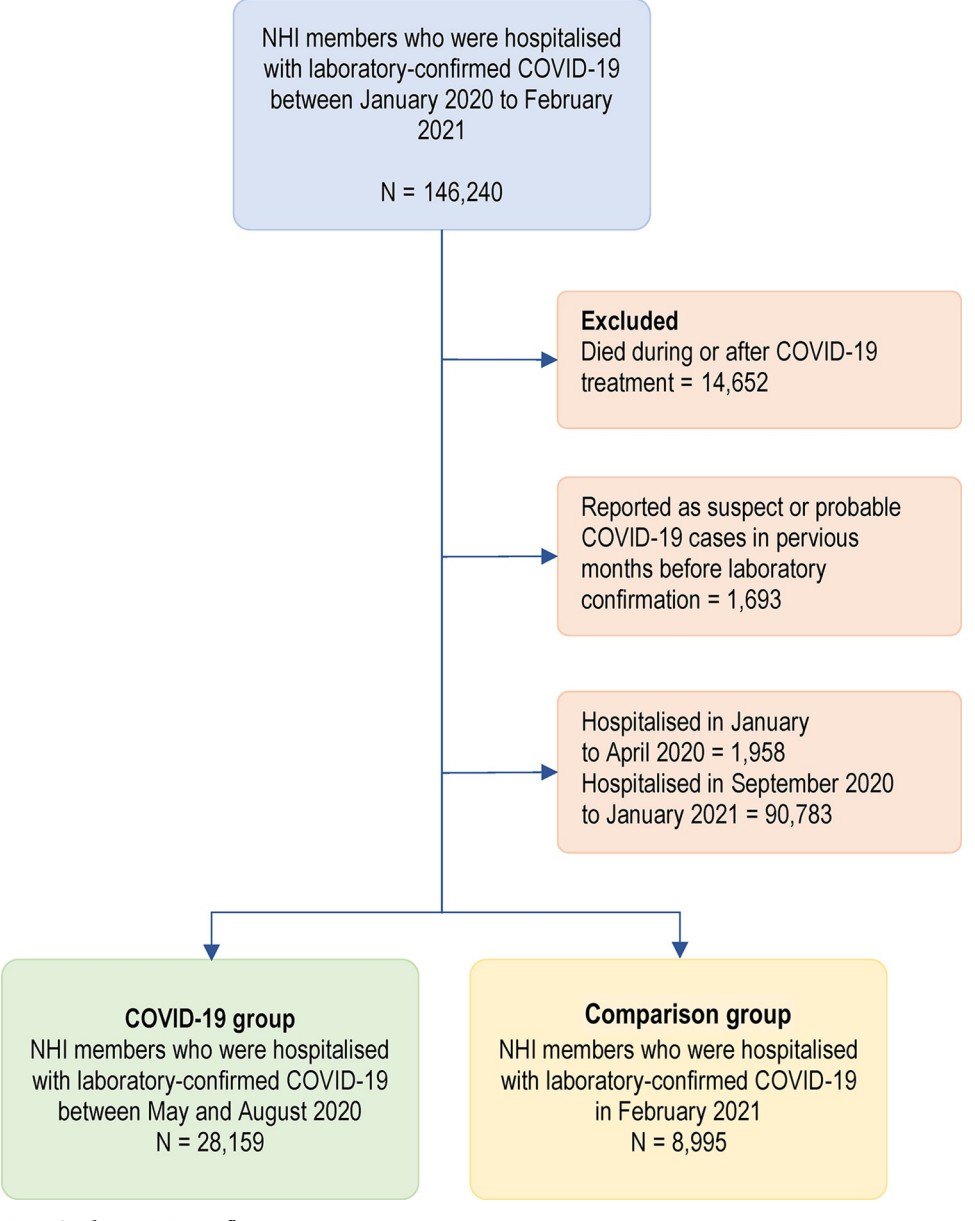

**Fig 2. Study participants flow.**

hospitalised as suspect or probable COVID-19 cases in the previous month before laboratory confirmation. Fig 2 depicts the flow of study participants and their classification into COVID-19 and comparison group. Our final sample consisted of 28,159 individuals in the COVID-19 group (those in hospitals with COVID-19 between May and August 2020) and 8,995 individuals in the comparison group (those in hospitals with COVID-19 in February 2021).

The average age of our sample (per January 2020) ranged from 44.2 to 46.5 years old, with some variations between groups. There were slightly more females (51.7% to 54.2%) and substantially higher individuals who were non-subsidised members of the NHI (81% to 85.9%). The subsidised scheme of the NHI, either provided by the central or local government, was intended to target the poorer population. Based on the distribution of severity level, which reflects the intensity of hospital resource utilization, most of the patients were hospitalized

**Table 1. Baseline characteristics of COVID-19 cases treated in hospitals in Indonesia from May to August 2020 compared with the comparison group.**

|  | COVID-19 group–Month in hospital with COVID-19 | | | | | Comparison group |
|  | May 20 | Jun. 20 | Jul. 20 | Aug. 20 | All |  |
|---|---|---|---|---|---|---|
| Age, mean (SD) | 44.6 (15.8) | 44.5 (15.8) | 44.2 (15.6) | 45.4 (15.6) | 44.7 (15.6) | 46.5 (16.2) |
| *Age group, %* |  |  |  |  |  |  |
| <20 | 4.3 | 4.1 | 4.2 | 3.8 | 4.0 | 4.3 |
| 20–39 | 34.8 | 34.6 | 34.7 | 32.4 | 33.9 | 29.9 |
| 40–59 | 42.3 | 42.4 | 44.5 | 45.1 | 44.1 | 42.9 |
| ≥40 | 18.6 | 18.8 | 16.6 | 18.8 | 18.1 | 22.9 |
| *Sex, %* |  |  |  |  |  |  |
| Male | 46.9 | 46.4 | 48.3 | 48.3 | 47.8 | 45.8 |
| Female | 53.1 | 53.6 | 51.7 | 51.7 | 52.2 | 54.2 |
| *Membership scheme, %* |  |  |  |  |  |  |
| Subsidised | 19.0 | 18.3 | 16.2 | 17.0 | 17.2 | 14.1 |
| Non-subsidized | 81.0 | 81.7 | 83.8 | 83.0 | 82.8 | 85.9 |
| *Severity, %* |  |  |  |  |  |  |
| None (outpatient) | 4.4 | 7.0 | 7.4 | 9.2 | 7.7 | 10.7 |
| Level 1 | 62.8 | 60.8 | 64.2 | 61.3 | 62.3 | 61.9 |
| Level 2 | 11.4 | 13.1 | 13.3 | 14.4 | 13.5 | 12.3 |
| Level 3 | 21.4 | 19.1 | 15.0 | 15.1 | 16.5 | 15.1 |
| *Comorbidity, %* |  |  |  |  |  |  |
| Hypertension | 11.1 | 12.7 | 11.6 | 12.7 | 12.2 | 14.6 |
| Diabetes | 10.9 | 11.8 | 11.2 | 12.3 | 11.7 | 13.0 |
| Heart disease | 6.2 | 7.2 | 6.4 | 7.4 | 6.9 | 8.2 |
| Tuberculosis | 2.3 | 2.1 | 1.8 | 1.8 | 1.9 | 2.3 |
| Asthma | 2.4 | 2.0 | 2.4 | 2.3 | 2.3 | 2.4 |
| COPD | 2.0 | 1.6 | 1.3 | 1.5 | 1.5 | 1.8 |
| Cancer | 4.1 | 4.4 | 3.9 | 4.4 | 4.2 | 4.6 |
| Liver disease | 1.8 | 1.4 | 1.3 | 1.3 | 1.4 | 1.6 |
| Kidney failure | 2.7 | 2.7 | 2.6 | 3.2 | 2.8 | 2.9 |
| *Provinces by island, %* |  |  |  |  |  |  |
| Sumatera | 13.8 | 9.2 | 9.6 | 12.6 | 11.0 | 8.7 |
| Java-Bali | 58.3 | 63.4 | 68.9 | 73.0 | 68.2 | 80.7 |
| Kalimantan | 7.3 | 7.7 | 7.5 | 7.4 | 7.5 | 4.6 |
| Sulawesi | 13.5 | 12.7 | 7.1 | 3.2 | 7.5 | 1.0 |
| Nusa Tenggara, Maluku, and Papua | 6.4 | 6.4 | 5.4 | 2.7 | 4.7 | 4.7 |
| Observations | 2,796 | 5,777 | 9,237 | 10,349 | 28,159 | 8,995 |

Note: COVID-19 group were divided into month based on their initial laboratory-confirmed diagnoses that were recorded in the COVID-19 hospital claims dataset. Individuals who died of any causes during or after COVID-19 treatment were excluded. Comparison group consisted of individuals who were diagnosed and treated in hospital with COVID-19 in February 2021. Individuals' age was in January 2020. Severity level is based on hospital resource utilisation during the first treatment episode where patients were laboratory-confirmed.

when they were confirmed with COVID-19 infection, although with decreasing trend (96% in May 2020 and 89% in the comparison group). Most of the study population were from Java and Bali provinces (58.2% for May 2020 group to 80.7% in the comparison group). Lastly, hypertension (11.1% to 14.6%) and diabetes (10.9% to 13%) were the two most common pre-COVID-19 comorbid–defined as diagnoses received by individuals before January 2020 (see Table 1).

## Changes in utilisation of hospital services

Fig 3 presented the difference-in-difference event-study estimates in outpatient visits rate of the COVID-19 group compared with the comparison group separately by treatment month. We presented the results for all ages, by age (<40 and ≥40 years old), and by gender. Our results indicated that up to the reference month, the differences in outcome between the COVID-19 and comparison group were close to zero and all statistically insignificant. This finding supported our assumption that the trend in outcome between the two groups were similar in the absence of COVID-19.

Starting from three months before treatment month, we observed some statistically significant increases in outpatient utilisation rate that peaked at the month of treatment with about 20 more visits per 100 individuals per month (p-value <0.001). In the subsequent months, those in hospitals with COVID-19 from June to August were still using more outpatient visits in about two to three months. A similar pattern was not observed for those who got COVID-19 in May. Further analysis indicated that this post-COVID trend was driven mainly by the older population. There were no substantial differences between males and female prior to month of treatment. However, the post-treatment trend indicated that males used more outpatient services after treated with COVID-19 (Fig 3).

As for inpatient utilisation, overall, we only observed a significant increase of about 21 to 33 additional admissions per 1,000 individuals (p < 0.001) at one month before and after COVID-19. There is no apparent gap between younger and older individuals or males and females with regards to inpatient utilisation as shown by the overlapping confidence interval in Fig 4.

## Changes in total costs of hospital services

Tables 2–4 presented the estimated difference-in-difference event-study in total costs of hospital services of the COVID-19 group compared with the comparison group, separately for each month when individuals were in hospitals with COVID-19. Results for all ages were shown in Table 2, while heterogeneity analyses by age and gender were provided in Tables 3 and 4, respectively.

In Table 2, we found a statistically significant increase of about 107 to 129 thousand Rupiahs in monthly hospital costs one month before the month of COVID-19 treatment (p<0.01), except for those who were treated in August 2020 with only 56 thousand Rupiahs increase (p<0.05). In addition, those who were treated in June and July seemed to have higher monthly hospital costs starting from three months (for those treated in July) or two months (for those treated in June). Compared to the difference at the reference time (month -4 to -6), we found no statistically significant gap in inpatient use in the previous months (month -7 to -9) between COVID-19 and the comparison group. This finding indicated that the two groups were relatively similar beyond six months before COVID-19 treatment.

During the month of treatment, the NHI expensed about 100 to 169 million Rupiahs more to those with COVID-19 than the comparison group (p<0.001). The figure was higher for individuals aged 40 years or older, with about 112 to 186 million Rupiahs increase in expenditure (p<0.001). Similarly, the cost was also higher among males with COVID-19 with average cost of 106 to 173 million rupiahs more compared to the comparison group. Afterwards, we still observed greater monthly hospital claims up to five months, with a decreased amount over time. The differences ranged between 1 to 2.5 million rupiahs in the first month (p<0.001) to about 84 to 224 thousand rupiahs in the fifth month after COVID-19 treatment (although it is only statistically significant with p<0.01 for those who were treated in June 2020) (see Table 2).

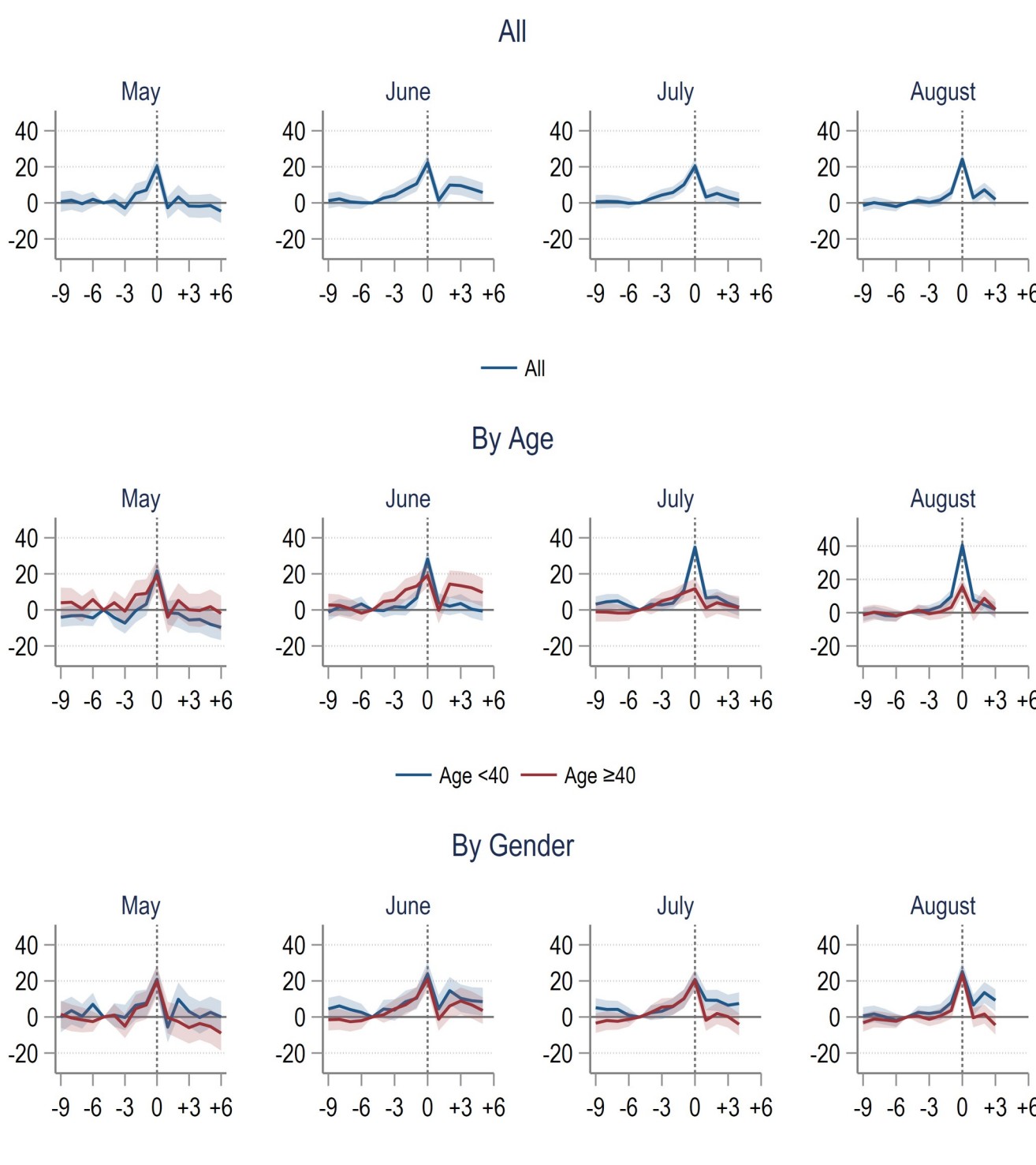

**Fig 3. Difference-in-difference estimates of hospital outpatient visit rate of COVID-19 group compared to comparison group.** Note: The outcome is the number of inpatient stays per 100 individuals per month. The horizontal axis represented months relative to COVID-19 hospital treatment. Lines represented the estimated difference between the COVID-19 and comparison groups, controlling for demographic characteristics. Shaded areas represented the 95% Confidence Intervals.

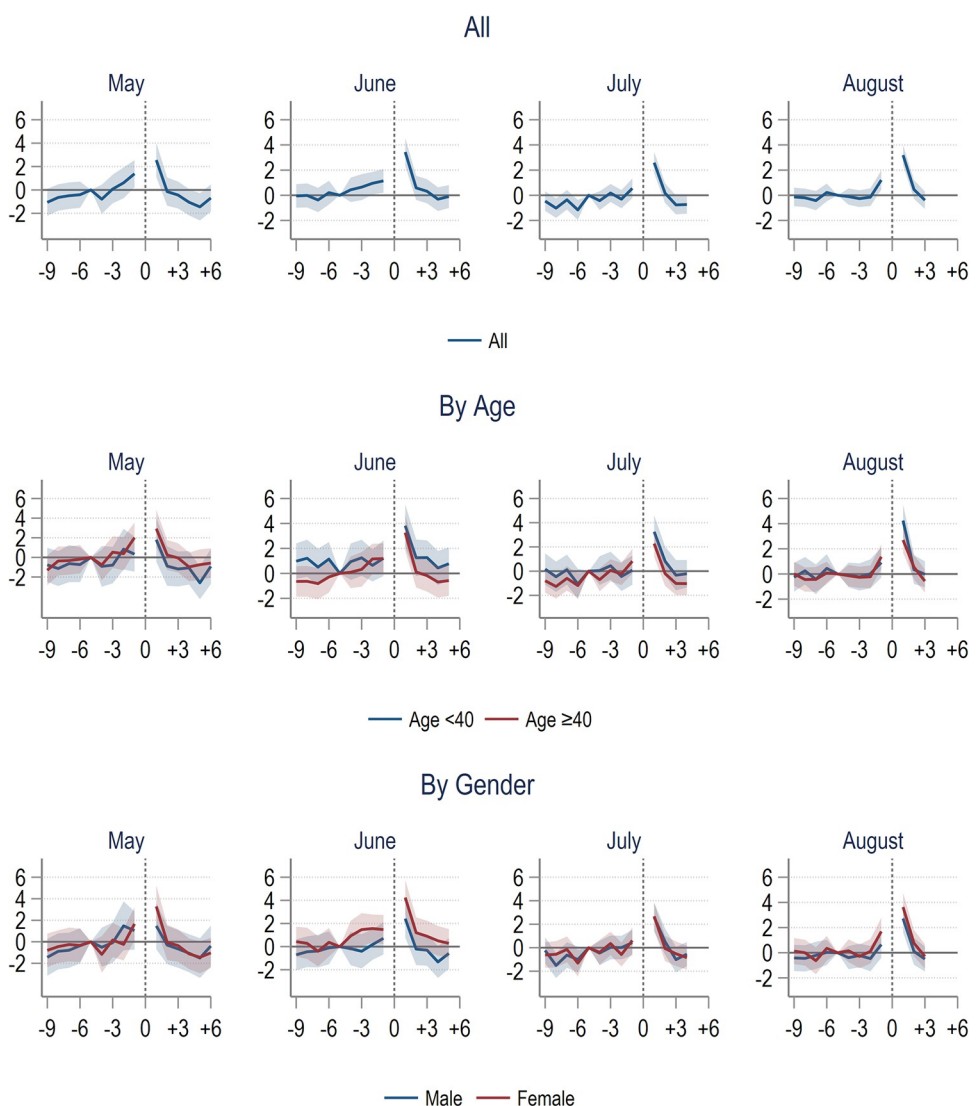

**Fig 4. Difference-in-difference estimates of hospital inpatient rate of COVID-19 group compared to comparison group.** Note: The outcome variable is the number of inpatient stays per 100 individuals per month. The horizontal axis represented months relative to COVID-19 hospital treatment. Lines represented the estimated difference between the COVID-19 and comparison groups, controlling for demographic characteristics. Shaded areas represented the 95% Confidence Intervals. Estimates for the month during laboratory-confirmed COVID-19 in hospital treatment were omitted because almost all individuals in the COVID-19 group had at least one hospital encounter.

Lastly, our analysis by age group revealed that the higher post-Covid hospital claims were concentrated in the older group (see Table 3). No substantial differences were found between males and females in post-Covid hospital claims (see Table 4).

In addition to the event-study estimates presented above, for transparency, we also reported the raw mean of monthly outpatient utilisation, inpatient utilisation, and hospital costs among COVID-19 groups and the comparison group (for all individuals) in Tables B1-B3 in S2 Table respectively.

**Table 2. Difference-in-difference estimates of total costs of hospital services of COVID-19 group compared to comparison group.**

|  | Month in hospital with COVID-19 | | | | | | | |
|---|---|---|---|---|---|---|---|---|
|  | May 20 | | Jun. 20 | | Jul. 20 | | Aug. 20 | |
| -7 to -9 month | -30.2 | (31.6) | -21.8 | (22.9) | -3.36 | (21.1) | -24.7 | (22.1) |
| -4 to -6 month | Ref. | | | | | | | |
| -3 month | 27.7 | (42.7) | 39.6 | (32) | 48.5* | (22.8) | 3.76 | (21.6) |
| -2 month | 32.2 | (44) | 76.2** | (27.5) | 22.8 | (20.2) | -4.99 | (21.8) |
| -1 month | 129** | (45.3) | 107** | (35.1) | 126*** | (35.6) | 56.1* | (27.6) |
| 0 (COVID-19) | 168,649*** | (2,259) | 136,843*** | (1,285) | 109,998*** | (860) | 99,732*** | (734) |
| +1 month | 2,453*** | (425) | 2,377*** | (313) | 1,507*** | (186) | 1,071*** | (116) |
| +2 month | 611*** | (178) | 286*** | (67.1) | 296*** | (65.1) | 310*** | (63.5) |
| +3 month | 277* | (139) | 357** | (111) | 133** | (47.4) | 102* | (41.8) |
| +4 month | 457* | (206) | 104* | (51.9) | 165** | (50.1) | | |
| +5 month | 84.1 | (79.7) | 224** | (73.6) | | | | |
| +6 month | 76.4 | (75.9) | | | | | | |
| Observations | 188,640 | | 221,565 | | 255,206 | | 251,459 | |
| Individuals | 11,790 | | 14,771 | | 18,229 | | 19,343 | |

Note: Coefficients represented the estimated differences in total costs of hospital services (in thousand Rupiahs) for both outpatient and inpatient and for all causes or diagnoses. Regressions control for individuals' year of birth, gender, COVID-19 severity, NHI membership segment, comorbidities prior to 2020 interacted with month indicator, as well as month and province fixed effects. Standard errors clustered at individual level in parentheses * $p < 0.05$, ** $p < 0.01$, *** $p < 0.001$.

## Discussion

### Main findings

Our results indicated some elevated uses and costs of hospital services months before and after the treatment month. Specifically, we found an increase in outpatient utilisation that happened three months before and up to three months after the treatment and an uptick in admissions rate one month before and after the treatment. These observed post-COVID trends were consistent with a previous study in England that found a higher readmission rate among discharged COVID-19 patients compared to comparison group [13]. Other studies reported that 20% of COVID-19 patients might require rehospitalisation, especially in older individuals and those with comorbidities [23–25]. This finding was consistent with our results where higher post-COVID utilisation was observed mainly in the older patients. In contrast, two studies in Norway and Denmark that only included mild cases did not find a short-term increase in specialist care [14, 15, 26].

Although multiple studies have reported that the presence of sequelae after SARS-CoV-2 infection might explain the elevated healthcare use in the post-treatment period [13, 27], other factors might still exist. The example includes the exacerbation of prior comorbidities or newly diagnosed conditions caused by extensive examinations during patients' acute COVID-19 treatment episodes. For the latter, our findings of a short-term increase in utilisation indicated that, if this situation happened, the conditions were unlikely to be chronic.

Another important result from this study was the increased hospital utilisation before the treatment month. One possibility is that people with recent medical conditions were more likely to have more severe symptoms, hence increasing the risk of requiring hospital treatment [28]. The second one is the limited testing capacity and clear clinical guide in the early phase of the pandemic in Indonesia. This situation could lead individuals to visit healthcare facilities multiple times before being treated as confirmed COVID-19 cases [29–31]. In addition, the increased use of healthcare and costs from five months before treatment could suggest that

**Table 3. Difference-in-difference estimates of total costs of hospital services of COVID-19 group compared to comparison group by age.**

| | Month in hospital with COVID-19 | | | | | | | |
|---|---|---|---|---|---|---|---|---|
| | May 20 | | Jun. 20 | | Jul. 20 | | Aug. 20 | |
| *Panel A. Age <40* | | | | | | | | |
| -7 to -9 month | -19.7 | (33.5) | 32.7 | (24.2) | 39.2 | (22.1) | -22 | (21.1) |
| -4 to -6 month | Ref. | | | | | | | |
| -3 month | -58.2 | (46.7) | 50.1 | (29.6) | 33 | (30.7) | -2.23 | (29.6) |
| -2 month | 42.9 | (48.8) | 5.18 | (34.3) | -3.75 | (26.8) | 11.9 | (26.4) |
| -1 month | 109 | (79.8) | 21.9 | (27.6) | 99.1 | (54.5) | 23.2 | (32.4) |
| **0 (COVID-19)** | 136,762*** | (3,435) | 107,241*** | (1,792) | 86,150*** | (1,161) | 76,387*** | (984) |
| +1 month | 1,197** | (369) | 2,012*** | (404) | 1,207*** | (207) | 780*** | (136) |
| +2 month | 200 | (166) | 274* | (107) | 289*** | (83.9) | 237** | (77.4) |
| +3 month | 128 | (115) | 331* | (151) | 167* | (78.8) | 79.6 | (61.1) |
| +4 month | 309 | (237) | 57.4 | (62.2) | 132 | (69.6) | | |
| +5 month | 84.3 | (153) | 380* | (172) | | | | |
| +6 month | -51.2 | (78.6) | | | | | | |
| Observations | 66,736 | | 79,710 | | 93,338 | | 88,582 | |
| Individuals | 4,171 | | 5,314 | | 6,667 | | 6,814 | |
| *Panel B. Age ≥ 40* | | | | | | | | |
| -7 to -9 month | -44.2 | (48) | -53.2 | (34) | -26.9 | (30.9) | -22.3 | (32.2) |
| -4 to -6 month | Ref. | | | | | | | |
| -3 month | 71.1 | (62.9) | 35.2 | (47.5) | 57.9 | (31.5) | 7.04 | (29.2) |
| -2 month | 21.7 | (65.1) | 117** | (38.8) | 35 | (28) | -15.4 | (30.1) |
| -1 month | 140* | (55.5) | 159** | (53) | 138** | (45.8) | 71.5 | (38.7) |
| **0 (COVID-19)** | 186,400*** | (2,910) | 154,274*** | (1,714) | 123,675*** | (1,153) | 112,049*** | (973) |
| +1 month | 3,176*** | (634) | 2,599*** | (440) | 1,677*** | (268) | 1,231*** | (163) |
| +2 month | 861** | (267) | 276*** | (82.1) | 291** | (89.1) | 352*** | (88.6) |
| +3 month | 356 | (207) | 375* | (153) | 105 | (58.5) | 116* | (55) |
| +4 month | 546 | (297) | 138 | (74.6) | 175** | (67.1) | | |
| +5 month | 89.3 | (91.9) | 136* | (58.3) | | | | |
| +6 month | 155 | (115) | | | | | | |
| Observations | 121,904 | | 141,855 | | 161,868 | | 162,877 | |
| Individuals | 7,619 | | 9,457 | | 11,562 | | 12,529 | |

Note: Coefficients represented the estimated differences in total costs of hospital services (in thousand Rupiahs) for both outpatient and inpatient and for all causes or diagnoses. Regressions control for individuals' year of birth, gender, COVID-19 severity, NHI membership segment, comorbidities prior to 2020 interacted with month indicator, as well as month and province fixed effects. Standard errors clustered at individual level in parentheses * $p < 0.05$, ** $p < 0.01$, *** $p < 0.001$.

patients with pre-existing conditions were more likely to experience severe COVID-19 outcomes, necessitating more extensive healthcare utilization [32].

This study also found a substantial variation with respect to the estimated changes in hospital costs across individuals treated in different calendar months. For example, during the month of treatment, patients treated in May 2020, on average, spent 169 million Rupiahs more than the comparison group. This figure decreased consistently with an estimated difference of about 138 million, 108 million, and 98 million Rupiahs among those treated in June, July, and August, respectively. Broadly, COVID-related claims in Indonesia during this period were based on a fixed per day payment scheme with values depending on the type of services (i.e. requiring ICU with or without a ventilator, negative pressure isolation room, or regular isolation room) and patient's comorbidities. In addition, the government also reimbursed hospital

**Table 4. Difference-in-difference estimates of total costs of hospital services of COVID-19 group compared to comparison group by gender.**

| | Month in hospital with COVID-19 | | | | | | | |
|---|---|---|---|---|---|---|---|---|
| | May 20 | | Jun. 20 | | Jul. 20 | | Aug. 20 | |
| *Panel A. Male* | | | | | | | | |
| -7 to -9 month | -58.6 | (56.3) | -34.9 | (39.6) | 1.82 | (35.2) | -51.8 | (38.3) |
| -4 to -6 month | Ref. | | | | | | | |
| -3 month | 11.2 | (71) | 24.4 | (47.9) | 52.3 | (37.1) | 45.9 | (39.1) |
| -2 month | 71.8 | (81.1) | 58.7 | (41.3) | 51.9 | (32) | -29 | (34.3) |
| -1 month | 132 | (84.5) | 119 | (60.9) | 175** | (63.9) | 19.9 | (45.1) |
| **0 (COVID-19)** | 173,020*** | (3,373) | 144,019*** | (1,956) | 116,517*** | (1,277) | 105,726*** | (1,105) |
| +1 month | 2,159*** | (513) | 2,294*** | (444) | 1,279*** | (188) | 1,186*** | (184) |
| +2 month | 718** | (251) | 333** | (112) | 333*** | (100) | 357*** | (105) |
| +3 month | 111 | (104) | 327* | (134) | 92.1 | (65.9) | 165* | (73) |
| +4 month | 256 | (158) | 3.1 | (61.7) | 217** | (76.8) | | |
| +5 month | 112 | (141) | 305* | (136) | | | | |
| +6 month | 168 | (137) | | | | | | |
| Observations | 86,832 | | 101,985 | | 120,064 | | 118,534 | |
| Individuals | 5,427 | | 6,799 | | 8,576 | | 9,118 | |
| *Panel B. Female* | | | | | | | | |
| -7 to -9 month | -10.1 | (34.2) | -10.1 | (25.5) | -5.03 | (24.7) | -.427 | (23.9) |
| -4 to -6 month | Ref. | | | | | | | |
| -3 month | 38.5 | (50.3) | 52.8 | (43) | 48.1 | (27.4) | -32.7 | (21.6) |
| -2 month | -3.14 | (44.9) | 89* | (36.2) | -1.86 | (25.7) | 16 | (27.1) |
| -1 month | 129** | (43.7) | 97.3* | (39.8) | 78.2* | (34.2) | 87.1* | (33.9) |
| **0 (COVID-19)** | 164,843*** | (3,050) | 130,447*** | (1,690) | 104,157*** | (1,160) | 94,272*** | (971) |
| +1 month | 2,694*** | (661) | 2,423*** | (429) | 1,723*** | (314) | 962*** | (145) |
| +2 month | 522* | (258) | 241** | (78.8) | 270** | (85.5) | 262*** | (73.6) |
| +3 month | 407 | (241) | 386* | (170) | 173* | (68.1) | 47.7 | (45.4) |
| +4 month | 627 | (354) | 189* | (80.6) | 126 | (66.9) | | |
| +5 month | 61.5 | (88.8) | 162* | (72.3) | | | | |
| +6 month | -26.2 | (67.8) | | | | | | |
| Observations | 101,808 | | 119,580 | | 135,142 | | 132,925 | |
| Individuals | 6,363 | | 7,972 | | 9,653 | | 10,225 | |

Note: Coefficients represented the estimated differences in total costs of hospital services (in thousand Rupiahs) for both outpatient and inpatient and for all causes or diagnoses. Regressions control for individuals' year of birth, gender, COVID-19 severity, NHI membership segment, comorbidities prior to 2020 interacted with month indicator, as well as month and province fixed effects. Standard errors clustered at individual level in parentheses * $p < 0.05$, ** $p < 0.01$, *** $p < 0.001$.

expenditure on personal protective equipment (PPE) [33, 34]. Therefore, any gaps in costs between individuals treated in different calendar months might reflect changes in hospital resource usage. One plausible reason for this pattern is the rapid development of clinical guidelines for COVID-19 management [35].

## Topics for future study

Given our results, a more detailed analysis of the type of diagnosis and procedure, including the use of primary care, is needed to fully understand the specific cause and type of increased healthcare utilisation in Indonesia. This type of question requires linked hospital and primary care utilisation, similar to what have been done in high-income countries [13, 14, 26, 27, 36].

The expanded database could also be utilised to examine the impact of less severe case of COVID-19 on healthcare utilisation.

In addition, since our study periods are mostly before the mass COVID-19 vaccination [37], a study using more recent data and a vaccination record is also needed. The observed patterns might change since the vaccination program is known to affect the course of COVID-19 disease. Lastly, due to the rapidly changing variants with different disease characteristics, data from different periods will be highly valuable in exploring the heterogeneity of its impact on healthcare utilisation.

## Strengths and limitations of this study

To our knowledge, this study was among the first in low-and-middle-income countries (LMICs) that used a national database to examine the impact of COVID-19 infection (as reported in hospital claims data) on healthcare utilisations. Compared to self-reported measure as usually collected in surveys, the use of an administrative database would minimise recall bias.

However, we are also aware that our study has several limitations. First, the study population was limited to NHI members who had at least one regular hospital encounter before being treated with laboratory-confirmed COVID-19. This made our sample smaller than the cumulative number of confirmed cases in Indonesia which were about 1.3 million cases as of February 2021. Second, our comparison group might be imperfect due to unmeasured time-varying factors that differentiate our COVID-19 group from the comparison group; hence residual confounding might still exist. For example, change in income or risk attitude–which was unmeasured in our dataset–might affect healthcare utilisation as well as the timing individuals were infected with COVID-19. We believe an improved dataset that covers all NHI members with more detailed individuals' characteristics might address some of these issues. Therefore, further cooperation with the Social Security Administrative Body of Health (BPJS-Kesehatan) and the Ministry of Health needs to be initiated. Third, our results around hospital costs are inherently sensitive to the change in clinical practice and the unit cost used for claim payments. Hence, our results should be carefully interpreted beyond the Indonesian context and the examined period. Lastly, due to the lack of information on symptoms and detailed resource utilisation, we could not identify whether patients developed Severe Acute Respiratory Infection (SARI) when treated with laboratory-confirmed COVID-19 infection.

## Policy implication

This study provided evidence of a short-term increase in utilisation of hospital services before and after COVID-19 hospitalisation in Indonesia that can be used for future health resource planning. This finding particularly important since the government of Indonesia is planning to shift the budget from the centrally allocated COVID-19 emergency fund to the regular NHI fund once the pandemic ends. As a social health insurance program that combines a subsidised scheme for the poor and a non-subsidised scheme based on a contributory mechanism for the non-poor (either through salary deduction or monthly premiums), it needs to anticipate the total cost of treating COVID-19 patients is required to keep its financial balance. For example, an additional government budget might be needed when a future outbreak arises and the estimated total costs of providing COVID-19 treatment exceed the available NHI fund.

## Conclusion

Compared to the comparison group, individuals with COVID-19 that required hospital treatment appeared to have a higher utilisation of outpatient and inpatient services a few months

before and after the month of treatment. This finding indicated that people who recently used more healthcare services were in a higher likelihood of contracting with COVID-19, either due to the worse health condition to begin with or because of the increasing contact with infected patients in the healthcare settings. The post-COVID pattern need to be considered by policy-makers when estimating the full impact of COVID-19 infection on healthcare resource utilisation, especially in more resource-constrained settings.

## Supporting information

**S1 Table. Summary of dataset used in the manuscript to create Figs 3 and 4.**
(PDF)

**S2 Table. Crude hospital utilisation rate and cost over time.**
(PDF)

## Acknowledgments

We thank Social Security Administrative Body of Health (BPJS-Kesehatan) team for data and anonymous reviewers for their constructive feedback and helped to improve this manuscript.

## Author Contributions

**Conceptualization:** Muhammad Fikru Rizal, Firdaus Hafidz, Gilbert Renardi Kusila, Wan Aisyiah, Dedy Revelino, Erzan Dhanalvin, Ilyasa, Citra Jaya, Benjamin Saut, Mahlil Ruby.

**Data curation:** Firdaus Hafidz, Wan Aisyiah, Dedy Revelino, Erzan Dhanalvin, Ayunda Oktavia, Ilyasa, Citra Jaya, Mahlil Ruby.

**Formal analysis:** Muhammad Fikru Rizal.

**Investigation:** Firdaus Hafidz, Wan Aisyiah.

**Methodology:** Muhammad Fikru Rizal, Firdaus Hafidz, Gilbert Renardi Kusila, Citra Jaya, Benjamin Saut.

**Resources:** Wan Aisyiah, Ayunda Oktavia, Ilyasa, Benjamin Saut, Mahlil Ruby.

**Supervision:** Citra Jaya, Benjamin Saut, Mahlil Ruby.

**Validation:** Firdaus Hafidz, Dedy Revelino.

**Writing – original draft:** Muhammad Fikru Rizal, Firdaus Hafidz, Gilbert Renardi Kusila.

**Writing – review & editing:** Muhammad Fikru Rizal, Firdaus Hafidz, Gilbert Renardi Kusila, Wan Aisyiah, Dedy Revelino, Erzan Dhanalvin, Ayunda Oktavia, Ilyasa, Citra Jaya, Benjamin Saut, Mahlil Ruby.

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
