## [Decision Letter · Decision Letter 0]

22 Mar 2023

PONE-D-22-31470Hospital services utilisation and cost before and after COVID-19 hospital treatment: evidence from IndonesiaPLOS ONE

Dear Dr. Firdaus Hafidz,

Thank you for submitting your manuscript to PLOS ONE. After careful consideration, we feel that it has merit but does not fully meet PLOS ONE’s publication criteria as it currently stands. Therefore, we invite you to submit a revised version of the manuscript that addresses the points raised during the review process.

We look forward to receiving your revised manuscript.

With Kind Regards,

Asst. Prof. Dr. Nemer Badwan

PhD in Economics and Finance

Assistant Professor of Economics and Finance

Academic Editor

PLOS ONE

Journal Requirements:

"This study is fully funded by Indonesia’s Social Security Administrative Body of Health (BPJS-Kesehatan). Award/grant number 456/BA/0621"

6. Please upload a new copy of Figure xxxx as the detail is not clear. Please follow the link for more information: https://blogs.plos.org/plos/2019/06/looking-good-tips-for-creating-your-plos-figures-graphics/ " https://blogs.plos.org/plos/2019/06/looking-good-tips-for-creating-your-plos-figures-graphics/.

Reviewers' comments:

Reviewer's Responses to Questions

**Comments to the Author**

1. Is the manuscript technically sound, and do the data support the conclusions?

Reviewer #1: Yes

Reviewer #2: Yes

2. Has the statistical analysis been performed appropriately and rigorously? 

Reviewer #1: Yes

Reviewer #2: Yes

3. Have the authors made all data underlying the findings in their manuscript fully available?

Reviewer #1: Yes

Reviewer #2: No

4. Is the manuscript presented in an intelligible fashion and written in standard English?

Reviewer #1: Yes

Reviewer #2: Yes

5. Review Comments to the Author

Reviewer #1: This paper presents important information on hospital services utilization and cost before and after COVID-19 hospital treatment. The article is well written but there is some minor editing required. Also, I have highlighted a few areas of concerns in the submitted manuscript (see uploaded attachment.

Reviewer #2: This is an informative and well written article. There are a number of suggestions for strengthening the content and findings.

1. In the introduction and discussion I would like to see more information on the population based epidemiology of covid-19 for Indonesia. This will help to better understand the landscape at this time. This study occurred during the first surges globally and the incidence and prevalence of the condition and other demographic characteristics of those effected and uneffected would be useful to see.

2. The data is based upon a large administrative data base from Indonesia subjects included "with 28,159 Indonesian NHI enrollees treated with laboratory-confirmed COVID-19, compared with 8,995 individuals never diagnosed with COVID-19 in 2020." The methods used for laboratory confirmed Covid-19 dx is not described in detail. Was this according to the WHO criteria for diagnosing Covid-19. In addition, as this was based upon national data, were the laboratory methods across the country uniform and standardized?

3. More detail is needed for understanding the Covid-19 positive and "never diagnosed" groups. What is the level of comparability for the two groups (covid positive and "never diagnosed" beingcompared). Perhaps a propensity analysis would be useful to better understand the distance for all of the sociodemographic variables included to best understand the differences between the two groups compared. When matching I would base this on the Mahalanobis distance (MD), with the selection of the caliper. The study is only as good as an indepth understanding of the control group.

4. For the control group, what is meant by "never diagnosed." Does this mean that those in the control group received the Covid-19 test and were negative? Is it possible that there was a fraction of individuals in the control group that were positive for covid?

4. An instrumental variable analysis might be useful to better understand unmeasured variables that go uncontrolled for in the analysis? What might such unmeasured variables be?

5. As this study was done during the earlier surges of the global pandemic it would be useful to understand the severity of the presenting covid-19 and if data is available have a breakdown of costs and hospitalizations for those severely impacted.

6. The focus for stratifications for hospital costs of services is by age (< 40 and >= 40). How was 40 years of age selected as the cutoff? Could more stratifications be included for age by a younger age group and the elderly? Results may be interesting. Can additional stratifications for other variables be included such as gender and educational level (a proxy for income).

7. I would like to see the Indonesian currency "Rupiahs" converted to US dollars. Or at least a footnote indicating the conversion rate.

6. PLOS authors have the option to publish the peer review history of their article (what does this mean?). If published, this will include your full peer review and any attached files.

Reviewer #1: **Yes: **Adebola Emmanuel Orimadegun

Reviewer #2: No

<quillbot-extension-portal></quillbot-extension-portal>

---

## [Author Response · Author response to Decision Letter 0]

20 May 2023

Dear Editor,

Thank you for considering our article for publication at PLOS One. We really appreciate the comments and suggestions from you and the reviewers. 

Editor

Journal Requirements:

and

 Thank you for the reminder. We have read the requirements and edited our files accordingly.

We have added in Methods section Ethical Clearance with the sentence as follows, 

“All data were fully anonymized before being accessed.”

"This study is fully funded by Indonesia’s Social Security Administrative Body of Health (BPJS-Kesehatan). Award/grant number 456/BA/0621"

We have put the additional information in the funding section as mentioned in the cover letter

"The funders had no role in study design, data collection and analysis, decision to publish, or preparation of the manuscript.”

We have moved the ethics statement to the Methods section.

We have updated our data availability statement to “All relevant data are within the manuscript and its Supporting Information files.” The relevant data are now submitted as supporting file.

6. Please upload a new copy of Figure xxxx as the detail is not clear. Please follow the link for more information: https://blogs.plos.org/plos/2019/06/looking-good-tips-for-creating-your-plos-figures-graphics/ " https://blogs.plos.org/plos/2019/06/looking-good-tips-for-creating-your-plos-figures-graphics/.

we have reuploaded our figures to follow the requirements with the maximum quality we could produce.

 

Reviewer 1

This paper presents important information on hospital services utilization and cost before and after COVID-19 hospital treatment. The article is well written but there is some minor editing required. Also, I have highlighted a few areas of concerns in the submitted manuscript (see uploaded attachment.

Thank you for the comments and suggestions. We have addressed all of them in the revised manuscript. Here the detail:

1. Introduction, Paragraph 1:

- We have replaced the sentence “A study in the US also mentioned that COVID-19 hospitalisation among patients with preexisting comorbidities was six times higher compared to those without ones” with a new sentence “A study conducted in Unites Stated found that COVID-19 hospitalization was six times higher in patients with preexisting comorbidities than in those without”

- We edited the last sentence and delete the sentence of “The same pattern also emerged from multiple settings, including in low-and-middle-income countries (LMICs),” 

2. Introduction, paragraph 2:

- We have deleted the word “also” in the first sentence

- We edited the sentence “Due to its multi-organ involvement, some COVID-19 symptoms can persist up to six months after infection remission or more, defined as long covid or post-covid syndrome” with a new sentence, “Due to its multi-organ involvement, some COVID-19 symptoms can persist up to six months after infection remission or more, which is defined as long covid or post-covid syndrome”

3. Methods, Section Study Population

We have added the comma after the word care in the sentence as follows 

“Individuals who were diagnosed based on RT-PCR positive results and treated, either in outpatient care or admitted to inpatient care, from May to August 2020 were included in the COVID-19 group.”

4. Figure 1’s Note

We have moved the note to the Study Period’s paragraph as follows, 

“We considered a period of August 2019 to November 2020 in our analysis. December 2020 and January 2021 were excluded due to the possibility of elevated healthcare utilisation two months before individuals in the control group were hospitalised with COVID-19 in February 2021. Hence, we had a maximum of six months (June 2020 to November 2020) and a minimum of three months (September to November 2020) post-COVID-19 follow-up. For pre-COVID period, we included nine months before COVID-19 hospital treatment. The grouping strategy and the period of observation used in our analyses is shown in Fig 1”

5. Figure 2’s Note

We have moved the note to the Study Participants’ paragraph as follows, 

“Figure 2 depicts the flow of study participants and their classification into COVID-19 and control group. “

6. Result, Section Changes in Utilisation of Hospital Services

We have deleted the sentence “The fifth month leading to the COVID-19 event was used as the reference period. We assumed the difference in outcome between the case and control in other periods to be constant” and put this explanation in the Methods section Study Population as follows:

“Those who treated in hospitals with COVID-19 in February 2021 were classified as the control group, assuming they never contracted COVID-19 in 2020. Note that in the NHI database, only results from RT-PCR (Reverse Transcriptase Polymerase Chain Reaction) done by Ministry of Health-accredited laboratories are used.” 

7. Results, Section Changes in Total Cost of Hospital Services

We have deleted the sentence, “In our regression, we used the period of 6 to 4 months before COVID-19 hospitalisation as the reference time where we assume no difference in admissions rate between the COVID-19 and the control group. We also lumped together 9 to 7 months before COVID19 to see whether there was any gap in outcomes between the two groups. “

The explanation related to the sentence has been included in the Methods, Section Study Period:

“We considered a period of August 2019 to November 2020 in our analysis. December 2020 and January 2021 were excluded due to the possibility of elevated healthcare utilisation two months before individuals in the control group were hospitalised with COVID-19 in February 2021. Hence, we had a maximum of six months (June 2020 to November 2020) and a minimum of three months (September to November 2020) post-COVID-19 follow-up. For pre-COVID period, we included nine months before COVID-19 hospital treatment. The grouping strategy and the period of observation used in our analyses is shown in Fig 1”

8. Discussion, Section Main Findings

- We have deleted this sentence, “Among 28,155 Indonesian National Health Insurance (NHI) enrollees who were in hospitals with laboratory-confirmed COVID-19 between May and August 2020”

- We revised the sentence “Specifically, we found an increase outpatient utilisation that happened three months before and up to three months after the treatment and an uptick in admissions rate one month before and after the treatment” with the additional word suggested by the reviewer as follows,

“Specifically, we found an increase in outpatient utilisation that happened three months before and up to three months after the treatment and an uptick in admissions rate one month before and after the treatment.”

- For the sentences related to the literature review, we have added some explanatory sentences to contrast their findings with ours, as follows

“Other studies reported that 20% of COVID-19 patients might require rehospitalisation, especially in older individuals and those with comorbidities [21-23]. This finding was consistent with our results where higher post-COVID utilisation was observed mainly in the older patients. In contrast, two studies in Norway and Denmark that only included mild cases did not find a short-term increase in specialist care [14, 15, 24].

9. Discussion, Section Policy Implication

We have replaced the word “elevation” with “to increase”

10. Conclusion

We have revised our conclusion not to recite our results. Therefore, our conclusion has been written as follows,

“Compared to the control group, individuals with COVID-19 that required hospital treatment appeared to have a higher utilisation of outpatient and inpatient services a few months before and after the month of treatment. This finding indicated that people who recently used more healthcare services were in a higher likelihood of contracting with COVID, either due to the worse health condition to begin with or because of the increasing contact with infected patients in the healthcare settings. More research is needed to fully understand the possibility of predicting COVID-19 infection based on prior healthcare utilisation. Lastly, the post-COVID pattern also need to be considered by policymakers when estimating the full impact of COVID-19 infection on healthcare resource utilisation, especially in more resource-constrained settings.”

 

Reviewer 2

This is an informative and well written article. There are a number of suggestions for strengthening the content and findings.

Thank you for your comments and suggestions. We have addressed all of them and provide details of our responses below.

1. In the introduction and discussion, I would like to see more information on the population based epidemiology of covid-19 for Indonesia. This will help to better understand the landscape at this time. This study occurred during the first surges globally and the incidence and prevalence of the condition and other demographic characteristics of those effected and unaffected would be useful to see.

In the introduction, we have added the following sentence to briefly lay out the epidemiology of COVID-19 in Indonesia, particularly in the earlier period.

“In Indonesia, about 12% of the hospitalised patients died in the first three months of the pandemic, with higher rate among older people, male, and those with multiple comorbidities [4]. A more recent study indicated that districts with higher proportion of elderly and lower healthcare capacity had higher rate of COVID-19 mortality [5].”

2. The data is based upon a large administrative data base from Indonesia subjects included "with 28,159 Indonesian NHI enrollees treated with laboratory-confirmed COVID-19, compared with 8,995 individuals never diagnosed with COVID-19 in 2020." The methods used for laboratory confirmed Covid-19 dx is not described in detail. Was this according to the WHO criteria for diagnosing Covid-19. In addition, as this was based upon national data, were the laboratory methods across the country uniform and standardized?

We have clarified this issue in our revised manuscript by adding the following sentence in our method section.

“Note that in the NHI database, only results from RT-PCR (Reverse Transcriptase Polymerase Chain Reaction) done by Ministry of Health-accredited laboratories are used.”

 

3. More detail is needed for understanding the Covid-19 positive and "never diagnosed" groups. What is the level of comparability for the two groups (covid positive and "never diagnosed" being compared). Perhaps a propensity analysis would be useful to better understand the distance for all of the sociodemographic variables included to best understand the differences between the two groups compared. When matching I would base this on the Mahalanobis distance (MD), with the selection of the caliper. The study is only as good as an indepth understanding of the control group.

We agree that the comparability in characteristics between the Covid-19 group and the “never diagnosed” or control group is important. We have shown in Table 1 (and explained in the Result section) that there are several characteristics differences. However, we believe that we have addressed these observed variations by including them as control variables in our regressions. We further highlight our approach by revising our statistical analysis subsection.

In our results, we have also shown that parallel pre-trends – which is one way to indicate that our DiD was valid – were mostly satisfied. Therefore, we decide not to proceed with matching on characteristics before employing the DiD estimation. For example, we have now mentioned 

“Our results indicated that up to the reference month, the differences in outcome between the COVID-19 and control group were close to zero and all statistically insignificant. This finding supported our assumption that the trend in outcome between the two groups were similar in the absence of COVID-19.”

4. For the control group, what is meant by "never diagnosed." Does this mean that those in the control group received the Covid-19 test and were negative? Is it possible that there was a fraction of individuals in the control group that were positive for covid?

The control group was the people that were diagnosed in February 2021, they 

---

## [Decision Letter · Decision Letter 1]

1 Sep 2023

PONE-D-22-31470R1Hospital services utilisation and cost before and after COVID-19 hospital treatment: evidence from IndonesiaPLOS ONE

Dear Dr. Firdaus Hafidz,

Thank you for submitting your manuscript to PLOS ONE. After careful consideration, we feel that it has merit but does not fully meet PLOS ONE’s publication criteria as it currently stands. Therefore, we invite you to submit a revised version of the manuscript that addresses the points raised during the review process.

We look forward to receiving your revised manuscript.

Kind Regards,

Asst. Prof. Dr. Nemer Badwan

PhD in Economics and Finance 

Assistant Professor of Economics and Finance

Academic Editor

PLOS ONE

Journal Requirements:

Reviewers' comments:

Reviewer's Responses to Questions

**Comments to the Author**

1. If the authors have adequately addressed your comments raised in a previous round of review and you feel that this manuscript is now acceptable for publication, you may indicate that here to bypass the “Comments to the Author” section, enter your conflict of interest statement in the “Confidential to Editor” section, and submit your "Accept" recommendation.

Reviewer #1: All comments have been addressed

Reviewer #3: (No Response)

2. Is the manuscript technically sound, and do the data support the conclusions?

Reviewer #1: Yes

Reviewer #3: Yes

3. Has the statistical analysis been performed appropriately and rigorously? 

Reviewer #1: Yes

Reviewer #3: Yes

4. Have the authors made all data underlying the findings in their manuscript fully available?

Reviewer #1: Yes

Reviewer #3: No

5. Is the manuscript presented in an intelligible fashion and written in standard English?

Reviewer #1: Yes

Reviewer #3: Yes

6. Review Comments to the Author

Reviewer #1: (No Response)

Reviewer #3: The authors conducted a retrospective study on parameters related to COVID-19 healthcare in Indonesia. The idea to focus on periods before or after treatment of COVID-19 patients is highly interesting. Scope and idea of the study are clearly described. The study is relevant for understanding the pandemic’s impact of the pandemic on the healthcare system.

Comments/questions:

1. I suppose “COVID-19” here is defined as a lab-confirmed SARS-CoV-2 infection. Are information available that allow for identifying patients who actually developed corresponding respiratory systems? What might be the influence of patients without SARI?

2. If “COVID-19” were indeed defined as SARS-CoV-2 infection, I would recommend including SARI as a comorbidity too.

3. How does hospital service utilization enter the statistical models? Is it a binary variable or a count per month? What kind of model was used, logistic or Poisson regression? Please add more details.

4. Is a linear model used for total claims (costs)? Did you check the distribution of costs? Was the variable transformed before entering the model? In my experience, this variable is highly skewed. Please add more details.

5. The model formula (line 146) contains two coefficients, alpha and beta. I suppose the regression estimated coefficients for the other covariates too. If this is true, please extend the formula.

6. The model formula appears to contain interaction terms between the month and several covariates (age, sex, comorbidities). Did the model also contain terms for main effects of these covariates?

7. I would like to know whether the authors did conduct the analyses stratified for total, age groups, and sex.

8. The authors present the proportions of different comorbidities. Are these comorbidities derived from ICD codes?

7. PLOS authors have the option to publish the peer review history of their article (what does this mean?). If published, this will include your full peer review and any attached files.

Reviewer #1: **Yes: **Adebola E. Orimadegun

Reviewer #3: No

While revising your submission, please upload your figure files to the Preflight Analysis and Conversion Engine (PACE) digital diagnostic tool, https://pacev2.apexcovantage.com/. PACE helps ensure that figures meet PLOS requirements. To use PACE, you must first register as a user. Registration is free. Then, login and navigate to the UPLOAD tab, where you will find detailed instructions on how to use the tool. If you encounter any issues or have any questions when using PACE, please email PLOS at figures@plos.org. Please note that Supporting Information files do not need this step.<quillbot-extension-portal></quillbot-extension-portal>

---

## [Author Response · Author response to Decision Letter 1]

15 Oct 2023

Dear Asst. Prof. Dr. Nemer Badwan,

I am writing to express my gratitude for the second valuable feedback provided by the reviewers and yourself regarding our submitted manuscript titled "Hospital services utilisation and cost before and after COVID-19 hospital treatment: evidence from Indonesia." Here below our details comments and response:  

Journal Requirements

We have checked all references, and found no retracted references. 

 

Reviewers' comments

Reviewer's Responses to Questions

Comments to the Author

1. If the authors have adequately addressed your comments raised in a previous round of review and you feel that this manuscript is now acceptable for publication, you may indicate that here to bypass the “Comments to the Author” section, enter your conflict of interest statement in the “Confidential to Editor” section, and submit your "Accept" recommendation.

Reviewer #1: All comments have been addressed.

Reviewer #3: (No Response)

2. Is the manuscript technically sound, and do the data support the conclusions?

Reviewer #1: Yes

Reviewer #3: Yes

3. Has the statistical analysis been performed appropriately and rigorously?

Reviewer #1: Yes

Reviewer #3: Yes

 

4. Have the authors made all data underlying the findings in their manuscript fully available?

Reviewer #1: Yes

Reviewer #3: No

We have revised our data sharing statement to the following. 

All relevant data used to produce the graphs in the paper are provided in S1 Table. The raw data that support the findings of this study are available from by Indonesia’s Social Security Administrative Body of Health (BPJS-Kesehatan), which were used under license for the current study, and so are not publicly available. Data is available from: https://data.bpjs-kesehatan.go.id/ by applying via the website. Other researchers will be able to access the data set in the same way as the authors.

5. Is the manuscript presented in an intelligible fashion and written in standard English?

Reviewer #1: Yes

Reviewer #3: Yes

6. Review Comments to the Author

Reviewer #1: (No Response)

Reviewer #3: 

Reviewer 3

The authors conducted a retrospective study on parameters related to COVID-19 healthcare in Indonesia. The idea to focus on periods before or after treatment of COVID-19 patients is highly interesting. Scope and idea of the study are clearly described. The study is relevant for understanding the pandemic’s impact of the pandemic on the healthcare system.

We thank the reviewer for the valuable comments. We have addressed each of your concerns below.

 I suppose “COVID-19” here is defined as a lab-confirmed SARS-CoV-2 infection. Are information available that allow for identifying patients who actually developed corresponding respiratory systems? What might be the influence of patients without SARI?

Thank you for raising the issue that there might be some laboratory-confirmed SARS-CoV-2 infections that did not develop into Severe Acute Respiratory Infection (SARI). Unfortunately, we do not have information on symptoms or detailed treatment received by the patients to identify SARI. Therefore, classifying COVID-19 cases with and without SARI is not possible. We have added this limitation to our revised manuscript.

“Lastly, due to the lack of information on symptoms and detailed resource utilisation, we could not identify whether patients developed Severe Acute Respiratory Infection (SARI) when treated with laboratory-confirmed COVID-19 infection.”

 If “COVID-19” were indeed defined as SARS-CoV-2 infection, I would recommend including SARI as a comorbidity too.

Because we cannot identify SARI, we cannot include it as comorbidity in our analysis. Similar to your first point, we have added this limitation to our revised manuscript.

 How does hospital service utilization enter the statistical models? Is it a binary variable or a count per month? What kind of model was used, logistic or Poisson regression? Please add more details.

Thank you for pointing out this issue. We are aware that the hospital service utilisation is a count data, representing the number of visits per individual per month. We have added the following sentence in our method section to clarify this concern:

“Although some of our outcomes were count variables, linear modelling can still be an appropriate choice, hence we estimate Equation 1 using Ordinary Least Square (OLS).”

Rothbard, S., Etheridge, J.C. & Murray, E.J. A Tutorial on Applying the Difference-in-Differences Method to Health Data. Curr Epidemiol Rep (2023). https://doi.org/10.1007/s40471-023-00327-x

 Is a linear model used for total claims (costs)? Did you check the distribution of costs? Was the variable transformed before entering the model? In my experience, this variable is highly skewed. Please add more details.

We agree that the skewness of claims (costs) data exist. However, since our aims is to estimate the quantity in absolute terms (Indonesian Rupiahs), we do not transform the costs variable first. To address this concern, we have added the following sentence to our method section.

“In addition, for costs outcome, we conducted additional analysis where the variable is first transformed into a log form because of the skewness of the data.”

We then described the findings in our result section.

“These results were similar when we used log-transformed value of the costs variables.”

 The model formula (line 146) contains two coefficients, alpha and beta. I suppose the regression estimated coefficients for the other covariates too. If this is true, please extend the formula.

We have extended the regression equation to clarify that we also estimated the coefficients of other covariates. The revised equation is written below:

Y_it=α_i+Σ_(t=Min)^(t=Max) 〖β_1〗_t Month_t×Covid_i+β_2 Covid_i+β_3 Month_t+〖β_4 X〗_i Month_t+〖β_5 X〗_i+ε_it

 The model formula appears to contain interaction terms between the month and several covariates (age, sex, comorbidities). Did the model also contain terms for main effects of these covariates?

Yes, it did. We have now clarified the confusion by expanding our regression equation, as we have mentioned in the point number 5. In this new equation, the coefficient for the main effects of the covariates were represented by β_5 vector. However, because these variables were used as control variables, we did not report and discuss the estimated coefficients further.

 I would like to know whether the authors did conduct the analyses stratified for total, age groups, and sex.

Yes, we did. We have already mentioned in the section “Statistical Analysis” of our initial manuscript as the following:

“Heterogeneity analysis was conducted by splitting the sample into younger (<40 years old) and older (≥40) population, as well as by gender.”

We have also reported the results for total sample, splitting by age group as well as gender or sex.

 The authors present the proportions of different comorbidities. Are these comorbidities derived from ICD codes?

ICD-10 codes of the primary and secondary diagnoses were used to define the comorbidities (see the list below)

Comorbidity ICD-10 Codes

Hypertension I10 – I16

Type 2 Diabetes Mellitus E11

Heart disease I11, I125

Tuberculosis A15 to A19

Asthma J45

COPD J44

Cancer C0 – C9; D1 – D4

Liver disease K70 – K77

Chronic Kidney Disease I12, I13, and N18

---

## [Editor Report · Decision Letter 2]

17 Oct 2023

PONE-D-22-31470R2Hospital services utilisation and cost before and after COVID-19 hospital treatment: evidence from IndonesiaPLOS ONE

Dear Dr. Hafidz,

Thank you for submitting your manuscript to PLOS ONE. After careful consideration, we feel that it has merit but does not fully meet PLOS ONE’s publication criteria as it currently stands. Therefore, we invite you to submit a revised version of the manuscript that addresses the points raised during the review process.

We look forward to receiving your revised manuscript.

Kind regards,

Asst. Prof. Dr. Nemer Badwan, Ph.D in Economics and Finance

Academic Editor

PLOS ONE
---

## [Author Response · Author response to Decision Letter 2]

17 Nov 2023

Dear Reviewer,

Thank you for your invaluable feedback on our second submission. We have carefully revised the manuscript in accordance with your suggestions. We noticed that we did not receive specific feedback on our third response; therefore, we have resubmitted it as it was. Regarding the references, we have thoroughly reviewed them and confirmed that none of the articles have been retracted. If there are any discrepancies in the reference list, please do let us know so we can make the necessary corrections.

Best regards,

Hafidz

---

## [Decision Letter · Decision Letter 3]

6 Dec 2023

PONE-D-22-31470R3Hospital services utilisation and cost before and after COVID-19 hospital treatment: evidence from IndonesiaPLOS ONE

Dear Dr. Hafidz,

Thank you for submitting your manuscript to PLOS ONE. After careful consideration, we feel that it has merit but does not fully meet PLOS ONE’s publication criteria as it currently stands. Therefore, we invite you to submit a revised version of the manuscript that addresses the points raised during the review process.

If applicable, we recommend that you deposit your laboratory protocols in protocols.io to enhance the reproducibility of your results. Protocols.io assigns your protocol its own identifier (DOI) so that it can be cited independently in the future. For instructions, see: https://journals.plos.org/plosone/s/submission-guidelines#loc-laboratory-protocols. Additionally, PLOS ONE offers an option for publishing peer-reviewed Lab Protocol articles, which describe protocols hosted on protocols.io. Read more information on sharing protocols at https://plos.org/protocols?utm_medium=editorial-email&utm_source=authorletters&utm_campaign=protocols.

We look forward to receiving your revised manuscript.

Kind Regards,

Asst. Prof. Dr. Nemer Badwan

Ph.D in Economics and Finance

Assistant Professor of Economics and Finance

Academic Editor

PLOS ONE

[Please do not edit.]

Reviewers' comments:

Reviewer's Responses to Questions

**Comments to the Author**

1. If the authors have adequately addressed your comments raised in a previous round of review and you feel that this manuscript is now acceptable for publication, you may indicate that here to bypass the “Comments to the Author” section, enter your conflict of interest statement in the “Confidential to Editor” section, and submit your "Accept" recommendation.

Reviewer #5: (No Response)

Reviewer #6: (No Response)

2. Is the manuscript technically sound, and do the data support the conclusions?

The manuscript must describe a technically sound piece of scientific research with data that supports the conclusions. Experiments must have been conducted rigorously with appropriate controls, replication, and sample sizes. The conclusions must be drawn appropriately based on the data presented. 

Reviewer #5: Partly

Reviewer #6: Partly

3. Has the statistical analysis been performed appropriately and rigorously? 

Reviewer #5: Yes

Reviewer #6: Yes

4. Have the authors made all the data underlying the findings in their manuscript fully available?

The PLOS Data policy requires authors to make all data underlying the findings described in their manuscript fully available without restriction, with a rare exception (please refer to the Data Availability Statement in the manuscript PDF file). The data should be provided as part of the manuscript or its supporting information or deposited to a public repository. For example, in addition to summary statistics, the data points behind means, medians and variance measures should be available. If there are restrictions on publicly sharing data—e.g., participant privacy or use of data from a third party—those must be specified.

Reviewer #5: Yes

Reviewer #6: No

5. Is the manuscript presented in an intelligible fashion and written in standard English?

Reviewer #5: Yes

Reviewer #6: Yes

6. Review Comments to the Author

Please use the space provided to explain your answers to the questions above. You may also include additional comments for the author, including concerns about dual publication, research ethics, or publication ethics. (Please upload your review as an attachment if it exceeds 20,000 characters.)

Reviewer #5: Major comments:

- Choice of reference period? What is the rationale for using the fifth month for hospital utilization and the 4-6 months before for cost? If you had chosen one month prior to hospitalization, the results would have been different. Why is the period in February used as comparison group?

- Is the aim of the study to investigate the utilization and cost for the ‘patients with COVID’ or the ‘COVID-19 disease’ itself? See the following text from your paper: “ We explored the short- to medium-term change in the pre- and post-acute period of COVID-19 treatment to fully understand the extent to which COVID-19 might be associated with the use of hospital services, especially in a resource-constrained setting.”

- I would like to include the crude rate of hospital utilization and cost, not only the estimated difference in the difference plot, to be sure of the assumption that the trend in outcome between the two groups was similar in the absence of COVID-19.

- I also want you to include the figures on the difference in the estimate for cost analysis and not only for health care utilization to see if the pre-trend assumption also holds here.

- In Figure 4 (in-patient, May period), I think the estimates of the difference are not zero in the months prior to the fifth month before COVID-19 treatment but rather increasing; this may be discussed in the Discussion section.

- Additionally, also you should also discuss what causes the increased use of health care and cost from 5 months before treatment; it is a long period, are these frail patients?

- And why are the patients older in the comparison group?

- There are more people in the comparison group with the non-subsidized membership scheme; could this affect health care utilization?

Minor comments:

- Please use the comparison group instead of the control group since it is not an RTC.

- Fig 1, misspelled in legend yellow, Months before COVID-10 hsopitalization

- Severity level: how are the different levels defined? Please include this information in the method section and in the table notation. Is this the same as the COVID-19 severity used as the control variable in the regression? And have all included patients with COVID-19 as their main cause of hospitalization, or could it also be that they were arbitrary was tested when admitted to the hospital?

- In figure 1, it is stated that "lines represented the estimated difference between the COVID-19 and 215 control groups, controlling for demographic characteristics," but the controlling variable is the same as stated in the tables: “Regressions control for individuals’ year of birth, gender, COVID-19 severity, NHI membership segment, comorbidities prior to 2020 interacted with the month indicator, as well as month and province fixed effects.” But in the data source section, the basic demographic characteristics included year of birth, sex, district and province, and type of NHI membership, but nothing about COVID-19 severity or comorbidity. Please indicate all controlling variables in the figure legends.

- Please be consistent with the word, like for instance, province or district instead of region, and COVID-severity instead of only severity.

- Use log-transformed values of the cost variables. What are the results from the additional analysis? I can’t see that Table 2 includes these results. “ These results were similar when we used the log-transformed value of the cost variables (see Table 2).”

Reviewer #6: The study authors conducted an analysis comparing inpatient hospital visit rates, outpatient hospital visit rates, and hospital-associated costs associated with COVID-19-positive patients compared with controls who have not yet contracted COVID-19. I believe the study is methodologically sound, but this can be strengthened with more details regarding data sources and the statistical methods.

- Page 3 line 62: “with” instead of “which” or some rewording of this first sentence is needed

- Page 6 line 128: “in in”

- Page 9 line 184: range of age minimum is 44.2, not 44.6, if I’m reading this correctly

- It took me a while, but after studying Figure 1 and looking at the results, I believe I understand the methods. Please confirm if I’m understanding this correctly. If I am, then I think the methods can be written differently to more simply explain what was done and why

o The control group (unexposed) contracted COVID-19 eventually. This fact is the reason why this group was chosen as a control (as a means to adjust for some confounders)

o The case group (exposed) contracted COVID-19 earlier (2020)

o The difference between exposed and unexposed was determined by time period (-9, -8, 0, 1, …, 6)

- We would expect, on average, that the cases and controls are similar with respect to demographic and clinical characteristics. However, this is generally not true as COVID-19 changed over time (e.g., virulence), immunity changed, and public health mandates changed as well. Standardized differences are not reported but are probably meaningfully different between cases and controls during the “pre” period. DiD methods often use propensity score methods to adjust for such factors. Why was this not considered? This is particularly of concern when, using age as an example

- It appears that there is a lack of parallel trends for the inpatient analysis (Figure 4).

- It’s unclear how clustering was taken into account since the same patient can be found in the analysis multiple times. From the table captions, it seems like there was some adjustment, but I did not see this mentioned in the methods.

- An assumption was made on page 6 that “those who were treated in hospitals with COVID-19 in February 2021 were classified as the control group, assuming they never contracted COVID-19 in 2020," but it’s not clear why an assumption had to be made. Rather, couldn’t this have been verified from the data?

- If coefficient 0.1t represents the month-specific DiD, is this what’s reported in Table 2? It isn’t clear from the table descriptor or the description of the results in the prose. For example, the paragraph beginning on page 17, line 270, does not mention interaction in the interpretation.

- It is not specified in the methods by which cases of COVID-19 were captured or the accuracy of this. Different countries may have had differential access to PCR testing over time, and inpatient testing may be more accurately documented than outpatient testing.

- The descriptors of cost were not presented. What types of costs were captured?

- I don’t think using prior medical history to predict COVID-19 infection should be mentioned in the conclusions.

7. PLOS authors have the option to publish the peer review history of their article (what does this mean?). If published, this will include your full peer review and any attached files.

If you choose “no”, your identity will remain anonymous, but your review may still be made public.

Reviewer #5: No

Reviewer #6: No

While revising your submission, please upload your figure files to the Preflight Analysis and Conversion Engine (PACE) digital diagnostic tool, https://pacev2.apexcovantage.com/. PACE helps ensure that figures meet PLOS requirements. To use PACE, you must first register as a user. Registration is free. Then, login and navigate to the UPLOAD tab, where you will find detailed instructions on how to use the tool. If you encounter any issues or have any questions when using PACE, please email PLOS at figures@plos.org. Please note that supporting information files do not need this step.

---

## [Author Response · Author response to Decision Letter 3]

11 Feb 2024

Dear Asst. Prof. Dr. Nemer Badwan,

I am writing to express my gratitude for the third valuable feedback provided by the reviewers and yourself regarding our submitted manuscript titled "Hospital services utilisation and cost before and after COVID-19 hospital treatment: evidence from Indonesia." Here below our details comments and response:  

Reviewers' comments

Reviewer #5

Major comments:

- Choice of reference period? What is the rationale for using the fifth month for hospital utilization and the 4-6 months before for cost? If you had chosen one month prior to hospitalization, the results would have been different. Why is the period in February used as comparison group?

This is an arbitrary decision. The point of reference period is to show that prior to the reference period, the differences in changes in outcomes over time is close to zero (the parallel pre-trend). This is a matter of presentation. Moreover, referencing Skyrud et al. (2021), no long-term elevation in healthcare use was observed in this period, supporting our timeframe choice for a balanced analysis.

February 2021 was selected as the control group to establish a COVID-19 baseline, excluding December 2020 and January 2021 due to potential elevated healthcare utilization before the comparison group were hospitalized. 

- Is the aim of the study to investigate the utilization and cost for the ‘patients with COVID’ or the ‘COVID-19 disease’ itself? See the following text from your paper: “ We explored the short- to medium-term change in the pre- and post-acute period of COVID-19 treatment to fully understand the extent to which COVID-19 might be associated with the use of hospital services, especially in a resource-constrained setting.”

It is for patients with COVID especially those who are treated in hospital, either inpatient or outpatient, due to COVID-9 which was confirmed based on standardised laboratory test. To clarify the aim, we revised the sentence: 

"Our study aims to examine the impact of COVID-19 by analysing both the direct effects on patients treated for the disease and the associated changes in hospital service utilization and costs. This dual perspective provides insights into the short- to medium-term adjustments within hospital services pre- and post-acute COVID-19 treatment periods, especially critical for understanding the pandemic's ramifications in resource-constrained settings."

- I would like to include the crude rate of hospital utilization and cost, not only the estimated difference in the difference plot, to be sure of the assumption that the trend in outcome between the two groups was similar in the absence of COVID-19.

Thank you for your suggestion. We have added Tables showing the crude rate of hospital utilisation and cost as supporting information in Table S2. 

- I also want you to include the figures on the difference in the estimate for cost analysis and not only for health care utilization to see if the pre-trend assumption also holds here.

Thank you for the suggestion. However, we believe this is unnecessary as our attempt to show the parallel pre-trend has been shown in Tables 2 to 4. Specifically, we have compared the total cost of COVID-19 and the comparison group at the reference period of 4 to 6 months before hospital treatment with a period of 7 to 9 months before. Hence, the estimates for the pre-reference period (7 to 9 months before hospital treatment), shown in the first row of the table, have indicated parallel pre-trends.

- In Figure 4 (in-patient, May period), I think the estimates of the difference are not zero in the months prior to the fifth month before COVID-19 treatment but rather increasing; this may be discussed in the Discussion section.

As discussed previously, the choice of reference group is a matter of presentation. We believe this does not require further discussion.

- Additionally, also you should also discuss what causes the increased use of health care and cost from 5 months before treatment; it is a long period, are these frail patients?

The increased use of healthcare and costs from 5 months before treatment could suggest that patients with pre-existing conditions are more likely to experience severe COVID-19 outcomes, necessitating more extensive healthcare utilization. This period might capture healthcare interactions related to managing chronic conditions, which are exacerbated or become more complex in the lead-up to a COVID-19 diagnosis. The notion of frail patients being more susceptible to higher healthcare use and costs aligns with broader medical literature, indicating that individuals with compromised health status face greater risks and healthcare needs during infectious disease outbreaks. We added in the discussion part

- And why are the patients older in the comparison group?

We are not sure about this, has not found any evidence support. One possible explanation for the older age profile in the comparison group (those affected in 2021) could be related to the evolution of the pandemic and changes in virus transmission dynamics over time. Initially, efforts might have focused on protecting and isolating older individuals, potentially leading to lower infection rates among this group in 2020. As the pandemic progressed into 2021, with the emergence of new variants or changes in public health measures, older individuals may have become more susceptible or exposed, leading to a higher representation in the comparison group.

- There are more people in the comparison group with the non-subsidized membership scheme; could this affect health care utilization?

Indeed, the presence of more individuals under the non-subsidized membership scheme in the comparison group could impact healthcare utilization patterns. However, it's important to note that during the pandemic, the government covered all patients' treatment costs for COVID-19, regardless of whether they had JKN (Indonesia's National Health Insurance) or not. This policy likely influenced a surge in access to COVID-19 services, with many claims categorized under the non-subsidized scheme. We anticipate this unique situation may have mitigated potential disparities in healthcare access between different insurance membership categories during the pandemic

Minor comments:

- Please use the comparison group instead of the control group since it is not an RTC.

Thanks, we replaced all control group to comparison group

- Fig 1, misspelled in legend yellow, Months before COVID-10 hospitalisation

Thanks, we revised the misspelling. 

- Severity level: how are the different levels defined? Please include this information in the method section and in the table notation. Is this the same as the COVID-19 severity used as the control variable in the regression? And have all included patients with COVID-19 as their main cause of hospitalization, or could it also be that they were arbitrary was tested when admitted to the hospital?

Severity levels in our study are defined based on the Indonesian Case-Based Groups (INA-CBGs) system, a case-mix payment system that utilizes a software grouper application. The severity level is influenced by complications and comorbidities, which are indicative of the resource intensity level required for treatment during the first treatment episode where patients were laboratory-confirmed. Yes, these severity levels are used as control variables in our regression analyses. All included patients had COVID-19 as their primary cause of hospitalization; the system ensures that the diagnosis is not arbitrary but based on rigorous clinical assessment upon admission.

We included the information in the method section and we had it in table notation 

- In figure 1, it is stated that "lines represented the estimated difference between the COVID-19 and 215 control groups, controlling for demographic characteristics," but the controlling variable is the same as stated in the tables: “Regressions control for individuals’ year of birth, gender, COVID-19 severity, NHI membership segment, comorbidities prior to 2020 interacted with the month indicator, as well as month and province fixed effects.” But in the data source section, the basic demographic characteristics included year of birth, sex, district and province, and type of NHI membership, but nothing about COVID-19 severity or comorbidity. Please indicate all controlling variables in the figure legends.

We included in the data source section. 

- Please be consistent with the word, like for instance, province or district instead of region, and COVID-severity instead of only severity.

Thanks, we revised the region into province

- Use log-transformed values of the cost variables. What are the results from the additional analysis? I can’t see that Table 2 includes these results. “ These results were similar when we used the log-transformed value of the cost variables (see Table 2).”

Apologies for any confusion caused regarding the log-transformation of cost variables. To clarify, the log-transformation of cost data was performed as part of our preliminary analysis to address data skewness, rather than as an additional, separate analysis. However, after tested it has similar results and easier to interpreted in non-log-transformed, ensuring a more accurate and meaningful statistical analysis. We revised the method section and deleted the results sentences. 

 

Reviewer #6

The study authors conducted an analysis comparing inpatient hospital visit rates, outpatient hospital visit rates, and hospital-associated costs associated with COVID-19-positive patients compared with controls who have not yet contracted COVID-19. I believe the study is methodologically sound, but this can be strengthened with more details regarding data sources and the statistical methods.

- Page 3 line 62: “with” instead of “which” or some rewording of this first sentence is needed

Replaced. Thanks,

- Page 6 line 128: “in in”

Replaced. Thanks,

- Page 9 line 184: range of age minimum is 44.2, not 44.6, if I’m reading this correctly

Replaced. Thanks. 

- It took me a while, but after studying Figure 1 and looking at the results, I believe I understand the methods. Please confirm if I’m understanding this correctly. If I am, then I think the methods can be written differently to more simply explain what was done and why

o The control group (unexposed) contracted COVID-19 eventually. This fact is the reason why this group was chosen as a control (as a means to adjust for some confounders)

o The case group (exposed) contracted COVID-19 earlier (2020)

o The difference between exposed and unexposed was determined by time period (-9, -8, 0, 1, …, 6)

Thanks for the suggestions. We revised in method section make it simpler way to explain.

- We would expect, on average, that the cases and controls are similar with respect to demographic and clinical characteristics. However, this is generally not true as COVID-19 changed over time (e.g., virulence), immunity changed, and public health mandates changed as well. Standardized differences are not reported but are probably meaningfully different between cases and controls during the “pre” period. DiD methods often use propensity score methods to adjust for such factors. Why was this not considered? This is particularly of concern when, using age as an example

We appreciate the reviewer's suggestion to employ propensity score methods (PSM) to adjust for differences between cases and controls. In anticipation of these concerns, we explored the application of PSM in our analysis. However, upon implementation, we encountered significant challenges: the confidence intervals of our estimates became notably wider, reducing the precision of our results. We think it is because the matching process necessitated a reduced sample size, potentially limiting the statistical power and generalizability of our findings. Moreover, we had already accounted for individual characteristics as control variables in our regression model, interacting these with the month variable to robustly address temporal variations in their effects on the outcomes. 

- It appears that there is a lack of parallel trends for the inpatient analysis (Figure 4).

Yes correct, we explained in the results section. 

- It’s unclear how clustering was taken into account since the same patient can be found in the analysis multiple times. From the table captions, it seems like there was some adjustment, but I did not see this mentioned in the methods.

Yes, it is true that the same patient may appear multiple times in our analysis due to the monthly clustering approach we adopted. As noted in our Tables, standard error were clustered at individual level. 

- An assumption was made on page 6 that “those who were treated in hospitals with COVID-19 in February 2021 were classified as the control group, assuming they never contracted COVID-19 in 2020," but it’s not clear why an assumption had to be made. Rather, couldn’t this have been verified from the data?

The assumption that individuals treated for COVID-19 in hospitals in February 2021 had not contracted the virus in 2020 was made to ensure that this is the right comparison group. While theoretically possible to verify past infections, this information is not available in our datasets.

- If coefficient 0.1t represents the month-specific DiD, is this what’s reported in Table 2? It isn’t clear from the table descriptor or the description of the results in the prose. For example, the paragraph beginning on page 17, line 270, does not mention interaction in the interpretation.

Not sure about the coefficient 0.1t you mentioned, but it the coefficients represented the estimated differences in total costs of hospital services (in thousand Rupiahs). Thanks we revised the description including paragraph page 17, line 270.

- It is not specified in the methods by which cases of COVID-19 were captured or the accuracy of this. Different countries may have had differential access to PCR testing over time, and inpatient testing may be more accurately documented than outpatient testing.

In our study, COVID-19 cases were identified through positive RT-PCR test results, with tests conducted by Ministry of Health-accredited laboratories. We acknowledge the potential variability in access to PCR testing across different regions and times, as well as the possibility that inpatient testing may be more systematically documented than outpatient testing.

- The descriptors of cost were not presented. What types of costs were captured?

Our analysis encompassed direct healthcare-related costs in bundled payment, such as hospitalization, medication, and diagnostic tests, outpatient visits and follow-up care at health facilities based on the claim reimbursement. We amended the manuscript to include a clear type of cost were analysed. 

- I don’t think using prior medical history to predict COVID-19 infection should be mentioned in the conclusions.

Thanks, deleted.

---

## [Decision Letter · Decision Letter 4]

29 Feb 2024

PONE-D-22-31470R4Hospital services utilization and cost before and after COVID-19 hospital treatment: evidence from IndonesiaPLOS ONE

Dear Dr. Hafidz,

Thank you for submitting your manuscript to PLOS ONE. After careful consideration, we feel that it has merit but does not fully meet PLOS ONE’s publication criteria as it currently stands. Therefore, we invite you to submit a revised version of the manuscript that addresses the points raised during the review process.

If applicable, we recommend that you deposit your laboratory protocols on protocols.io to enhance the reproducibility of your results. Protocols.io assigns your protocol its own identifier (DOI) so that it can be cited independently in the future. For instructions, see: https://journals.plos.org/plosone/s/submission-guidelines#loc-laboratory-protocols. Additionally, PLOS ONE offers an option for publishing peer-reviewed Lab Protocol articles, which describe protocols hosted on protocols.io. Read more information on sharing protocols at https://plos.org/protocols?utm_medium=editorial-email&utm_source=authorletters&utm_campaign=protocols.

We look forward to receiving your revised manuscript.

Kind Regards,

Asst. Prof. Dr. Nemer Badwan 

Ph.D in Economics and Finance

Assistant Professor of Economics and Finance

Academic Editor and Reviewer 

PLOS ONE

. [Please do not edit.]

Reviewers' comments:

Reviewer's Responses to Questions

**Comments to the Author**

1. If the authors have adequately addressed your comments raised in a previous round of review and you feel that this manuscript is now acceptable for publication, you may indicate that here to bypass the “Comments to the Author” section, enter your conflict of interest statement in the “Confidential to Editor” section, and submit your "Accept" recommendation.

Reviewer #5: All comments have been addressed

Reviewer #7: (No Response)

2. Is the manuscript technically sound, and do the data support the conclusions?

The manuscript must describe a technically sound piece of scientific research with data that supports the conclusions. Experiments must have been conducted rigorously with appropriate controls, replication, and sample sizes. The conclusions must be drawn appropriately based on the data presented. 

Reviewer #5: Yes

Reviewer #7: Partly

3. Has the statistical analysis been performed appropriately and rigorously? 

Reviewer #5: Yes

Reviewer #7: Yes

4. Have the authors made all the data underlying the findings in their manuscript fully available?

The PLOS Data policy requires authors to make all data underlying the findings described in their manuscript fully available without restriction, with a rare exception (please refer to the Data Availability Statement in the manuscript PDF file). The data should be provided as part of the manuscript or its supporting information or deposited in a public repository. For example, in addition to summary statistics, the data points behind means, medians and variance measures should be available. If there are restrictions on publicly sharing data—e.g., participant privacy or use of data from a third party—those must be specified.

Reviewer #5: Yes

Reviewer #7: No

5. Is the manuscript presented in an intelligible fashion and written in standard English?

Reviewer #5: Yes

Reviewer #7: Yes

6. Review Comments to the Author

Please use the space provided to explain your answers to the questions above. You may also include additional comments for the author, including concerns about dual publication, research ethics, or publication ethics. (Please upload your review as an attachment if it exceeds 20,000 characters.)

Reviewer #5: Thanks for the answers to all my previous comments. I only have one additional comment.

Could you refer to all the S2 tables in the manuscript?

Reviewer #7: Review report on PONE-D-22-31470_R4: Hospital services utilization and cost before and after COVID-19 hospital treatment: evidence from Indonesia

The authors estimate health care utilization before and after PCR-confirmed SARS-CoV-2 in Indonesia in 2020 by contrasting the utilization of patients with COVID-19 in 2020 to the utilization of patients with COVID-19 in February 2021. They find elevated utilization around the time of infection, but no difference after four months.

The paper is well-written, and the statistical analyses appear well-executed. As noted by the authors, it is important to undertake studies using registry data in LMIC too, and this is a fine example thereof. Still, I believe the study suffers from a couple of fundamental concerns that need to be discussed more carefully and sincerely before the paper can be accepted for publication.

The construction of the treatment (covid-19) and comparison (not yet covid-19) is crucial for the interpretation of the results. The author states that “the comparison group comprised individuals treated for COVID-19 in hospitals in February 2021, under the presumption they had not contracted the virus in 2020” (line 115). This implies that what is estimated is not the impact (on subsequent utilization/costs) of having COVID-19 vs. not having COVID-19 in 2020, but the impact (on subsequent utilization/costs) of having COVID-19 in 2020 vs. having COVID-19 in 2021. This distinction has important implications for interpretation. For example, the authors’ DiD estimates how much higher the costs of treating a COVID-19 patient were in 2020 compared to 2021 (not the costs of treating a COVID-19 patient vs. not treating one). And, as another example, the authors do not shed any light on the presence of sequelae after SARS-CoV-2 infection, only on the question of whether possible sequelae are different in 2020 vs. 2021.

Clearly, these are two different questions (i. effects on utilization of COVID; ii. effects on utilization of COVID in 2020 vs. 2021), due, e.g., to the strain on health care services and other societal restrictions at the beginning of the pandemic. It seems obvious that the services’ ability to treat patients (both those with COVID-19 and other patients) was very different in 2020 than in 2021 (and definitely in 2024). It is crucial that the authors make this distinction between the two research questions clear to the readers (including in the abstract) and that they improve precision in how they describe their research question (and findings) throughout the paper (including considering making the title communicate this better).

In doing so, the authors may consider describing briefly other ways of constructing the comparison group and its implications for the research question, e.g., using contemporaneous patients with negative PCR tests (in 2020), like in reference (14), which would address the question of the impact of SARS-CoV-2 vs. no SARS-CoV-2 (instead of SARS-CoV-2 in 2020 vs. 2021). The authors should also be careful in pointing out the differences in research questions when comparing them with previous studies and when discussing policy implications (both in Discussion). This said, it appears to me that the question actually addressed by the authors' regressions (impact on utilization of COVID in 2020 vs. 2021) is of interest; it just needs to be stated and motivated clearly to the reader.

It seems that those dying are excluded from the sample, instead of the more common approach of censuring them from the month of death: “Similarly, those who died during COVID-19 hospitalization or in subsequent months were also excluded from the analysis.” (line 114). If medical treatment for COVID-19 was more effective in 2021 than in 2020 (e.g., due to less strain on the services or some immunity due to low-dose exposure or vaccination (too early for that in Indonesia?)), this introduces a potentially serious bias to the analysis: The comparison group receives more utilization because they survive longer (which would lead to an underestimation of how much higher costs were in 2020 than in 2021). This problem is not easy to handle, but it absolutely deserves a serious and sincere discussion. It would also be important to inform the reader of the survival in the two groups (i.e., by providing death rates by 3-6 months or even the impact on death using the same regressions). Was survival measured for the same length of time in the comparison group?

Minor things.

I do not understand why “individuals who were under monitoring and recorded as suspect or probable cases of COVID-19 in the previous months before laboratory confirmation were excluded” (line 113), since it makes the relevance of the results restricted to those who are actually being tested (which I presume could depend on the severity of the infection or socioeconomic status, or, as noted by the authors, that the hospital/patient have access to PCRs done by a Ministry of Health-accredited laboratory). On the other hand, I do not see that this has important methodological problems (except that PCR testing capacity was presumably smaller relative to the number of infections in 2020 than in February 2021) as long as the reader understands that the population under study is narrow. However, in note to Table 1, it is stated that “individuals without any hospital encounters before January 2020 (..) were excluded.” How is this incorporated for the comparison group in a way that does not introduce differences/bias across the two groups?

It would be very informative if Table 1 also included the outcome variables (i.e., the raw out- and inpatient rates and costs by, e.g., 3 months) and a reference to Supplement Table 2 for details (but this supplement table is not informative about the comparison group since we are only informed about post-infection months for the treated (not the comparison)—please fix).

Eq. 1 includes individual fixed effect (alpha i), but also a dummy for being in the treatment group (Covid i). I presume there is a typo here, as the individual fixed effect would absorb all other time-invariant individual characteristics (and, obviously, being in the treatment group does not change over time).

Given the vast population of Indonesia, some information on why there are so few people in the sample is warranted.

I don’t understand why the cost results are not also presented in a figure (like the nice figures for in- and outpatient results; please refer the reader to Supplement Table 1 in the note to these figures).

I was not able to reach the data application site using the link in the data sharing statement.

“before” is lacking in line 31 of the abstract.

Strictly speaking, the PCR test is for SARS-CoV-2 (i.e., the virus) and not for COVID-19 (the disease); the authors might consider their precision in the use of these two terms.

7. PLOS authors have the option to publish the peer review history of their article (what does this mean?). If published, this will include your full peer review and any attached files.

If you choose "no," your identity will remain anonymous, but your review may still be made public.

Reviewer #5: No

Reviewer #7: No

While revising your submission, please upload your figure files to the Preflight Analysis and Conversion Engine (PACE) digital diagnostic tool, https://pacev2.apexcovantage.com/. PACE helps ensure that figures meet PLOS requirements. To use PACE, you must first register as a user. Registration is free. Then, login and navigate to the UPLOAD tab, where you will find detailed instructions on how to use the tool. If you encounter any issues or have any questions when using PACE, please email PLOS at figures@plos.org. Please note that supporting information files do not need this step.

---

## [Author Response · Author response to Decision Letter 4]

22 Mar 2024

Reviewer 5

Thanks for the answers to all my previous comments. I only have one additional comment.

Could you refer to all the S2 tables in the manuscript?

We have now added the following sentences to refer to the Supplement S2 Tables in the end of our Results Section.

“In addition to the event-study estimates presented above, for transparency, we also reported the raw mean of monthly outpatient utilisation, inpatient utilisation, and hospital costs among COVID-19 groups and the comparison group (for all individuals) in Supplement S2 Tables B1, B2, and B3, respectively.”

 

Reviewer 7

The construction of the treatment (covid-19) and comparison (not yet covid-19) is crucial for the interpretation of the results. The author states that “the comparison group comprised individuals treated for COVID-19 in hospitals in February 2021, under the presumption they had not contracted the virus in 2020” (line 115). This implies that what is estimated is not the impact (on subsequent utilization/costs) of having COVID-19 vs. not having COVID-19 in 2020, but the impact (on subsequent utilization/costs) of having COVID-19 in 2020 vs. having COVID-19 in 2021. This distinction has important implications for interpretation. For example, the authors’ DiD estimates how much higher the costs of treating a COVID-19 patient were in 2020 compared to 2021 (not the costs of treating a COVID-19 patient vs. not treating one). And, as another example, the authors do not shed any light on the presence of sequelae after SARS-CoV-2 infection, only on the question of whether possible sequelae are different in 2020 vs. 2021.

Clearly, these are two different questions (i. effects on utilization of COVID; ii. effects on utilization of COVID in 2020 vs. 2021), due, e.g., to the strain on health care services and other societal restrictions at the beginning of the pandemic. It seems obvious that the services’ ability to treat patients (both those with COVID-19 and other patients) was very different in 2020 than in 2021 (and definitely in 2024). It is crucial that the authors make this distinction between the two research questions clear to the readers (including in the abstract) and that they improve precision in how they describe their research question (and findings) throughout the paper (including considering making the title communicate this better).

In doing so, the authors may consider describing briefly other ways of constructing the comparison group and its implications for the research question, e.g., using contemporaneous patients with negative PCR tests (in 2020), like in reference (14), which would address the question of the impact of SARS-CoV-2 vs. no SARS-CoV-2 (instead of SARS-CoV-2 in 2020 vs. 2021). The authors should also be careful in pointing out the differences in research questions when comparing them with previous studies and when discussing policy implications (both in Discussion). This said, it appears to me that the question actually addressed by the authors' regressions (impact on utilization of COVID in 2020 vs. 2021) is of interest; it just needs to be stated and motivated clearly to the reader.

Thank you for your comment. In our Methods, we have made clear that the Comparison group, those who were treated with COVID-19 in February 2021 were assumed to not contracted COVID-19 in 2020. In the Study Population subsection, we mentioned “Those who treated in hospitals with COVID-19 in February 2021 were classified as the control group, assuming they never contracted COVID-19 in 2020.”

We have added the following sentence to elaborate our assumption.

"This assumption is plausible given reinfection rate within one year follow up was only about 5% [17].”

In Study Population subsection, we revised our explanation to address your concern.

“We considered a period of August 2019 to November 2020 in our analysis. During this period, the comparison group, those who were treated with COVID-19 in February 2021, was assumed to not yet contracted the disease."

We believe that this is now clear that in our post-COVID follow-up, our comparison group was realistically assumed to have not contracted COVID-19, hence appropriate as control group and our main research question and subsequent interpretation of the results remain. 

It seems that those dying are excluded from the sample, instead of the more common approach of censuring them from the month of death: “Similarly, those who died during COVID-19 hospitalization or in subsequent months were also excluded from the analysis.” (line 114). If medical treatment for COVID-19 was more effective in 2021 than in 2020 (e.g., due to less strain on the services or some immunity due to low-dose exposure or vaccination (too early for that in Indonesia?)), this introduces a potentially serious bias to the analysis: The comparison group receives more utilization because they survive longer (which would lead to an underestimation of how much higher costs were in 2020 than in 2021). This problem is not easy to handle, but it absolutely deserves a serious and sincere discussion. It would also be important to inform the reader of the survival in the two groups (i.e., by providing death rates by 3-6 months or even the impact on death using the same regressions). Was survival measured for the same length of time in the comparison group?

Thank you for your comment. First, we would like to reiterate that the comparison group (who were treated with COVID-19 in February 2021) had not yet contracted COVID during our pre- and post-COVID periods among those treated with COVID-19 in May to August 2020 (the COVID-19 group). This means that during these follow-up periods, the comparison group was assumed to use healthcare services that were not associated with them having contracted COVID-19. 

Second, our decision to focus on survivors was precisely to avoid the issue of reduced healthcare use of COVID-19 patients due to their higher risk of mortality following their hospital treatment. We added the following explanation to our Study Population subsection.

“This decision was made to focus on COVID-19 survivors, hence avoiding potential issue of reduced healthcare use by the COVID-19 group due to their higher post-hospital treatment mortality.”

 

Minor things.

I do not understand why “individuals who were under monitoring and recorded as suspect or probable cases of COVID-19 in the previous months before laboratory confirmation were excluded” (line 113), since it makes the relevance of the results restricted to those who are actually being tested (which I presume could depend on the severity of the infection or socioeconomic status, or, as noted by the authors, that the hospital/patient have access to PCRs done by a Ministry of Health-accredited laboratory). On the other hand, I do not see that this has important methodological problems (except that PCR testing capacity was presumably smaller relative to the number of infections in 2020 than in February 2021) as long as the reader understands that the population under study is narrow. However, in note to Table 1, it is stated that “individuals without any hospital encounters before January 2020 (..) were excluded.” How is this incorporated for the comparison group in a way that does not introduce differences/bias across the two groups?

We agree with your concern that this is part of the data limitations. We have mentioned that the narrowness of this study population in our study limitation subsection.

Regarding the exclusion of individuals without any hospital encounters before January 2020, we made a mistake in the note from previous iteration of the analysis. The current results did not make this exclusion. We have revised our note to Table 1 as the following.

“Individuals who died of any causes during or after COVID-19 treatment were excluded.”

It would be very informative if Table 1 also included the outcome variables (i.e., the raw out- and inpatient rates and costs by, e.g., 3 months) and a reference to Supplement Table 2 for details (but this supplement table is not informative about the comparison group since we are only informed about post-infection months for the treated (not the comparison)—please fix).

It would be difficult to provide the raw mean of our outcome variables in Table 1 for both COVID-19 and comparison group since the months that we used as pre- and post-COVID periods were different. In Figure 1, we have made it clear that, for example, in COVID-19 group who were hospitalised in May 2020, the post-COVID period that was analysed was June to November 2020. 

The comparison of raw outcome means for COVID-19 groups and the comparison group were provided in S2 Tables. To make the Tables more informative, we have now revised the Table notes as the following.

“The shaded cells are the post-COVID period for each COVID-19 group. The underlined figures are the reference period for each COVID-19 group. A simple analysis is to compare the outcomes in the same months of the COVID-19 groups with the comparison group, adjusted with the difference in outcome between the two groups at the reference month (the underlined figure). For example, the post-COVID period for May 2020 COVID-19 group is June to November 2020 and the reference month is December 2019. Robust standard errors in parentheses.” 

Eq. 1 includes individual fixed effect (alpha i), but also a dummy for being in the treatment group (Covid i). I presume there is a typo here, as the individual fixed effect would absorb all other time-invariant individual characteristics (and, obviously, being in the treatment group does not change over time).

Thank you for the correction. We have now deleted α_i from the equation since it is not estimated in our regression. 

Given the vast population of Indonesia, some information on why there are so few people in the sample is warranted.

We have now revised the first limitation of the paper as the following.

“First, the study population was limited to NHI members who had at least one regular hospital encounter before being treated with laboratory-confirmed COVID-19. This made our sample smaller than the cumulative number of confirmed cases in Indonesia which were about 1.3 million cases as of February 2021.”

I don’t understand why the cost results are not also presented in a figure (like the nice figures for in- and outpatient results; please refer the reader to Supplement Table 1 in the note to these figures).

We did not present the cost results as Figures because the scale of the estimated effect of COVID-19 across period greatly differed. For example, in Table 2, the point estimate for May 2020 COVID-19 group at 1 month after COVID-19 hospitalisation was 2,453 (in thousand IDR) while the point estimate at the hospitalisation month was 168,649 (in thousand IDR). Because of this concern, presenting the results as Figures would not be as informative as detailed results provided in Tables 2-4.

I was not able to reach the data application site using the link in the data sharing statement.

We are aware about the difficulty of accessing BPJS Kesehatan website from overseas. This issue is, however, beyond our capacity as researchers.

“before” is lacking in line 31 of the abstract.

Thank you for pointing this out. We have now revised this.

Strictly speaking, the PCR test is for SARS-CoV-2 (i.e., the virus) and not for COVID-19 (the disease); the authors might consider their precision in the use of these two terms.

Thank you for the suggestion. We have clearly stated in our Study Population section that we define COVID-19 in our paper as PCR-confirmed COVID-19 cases. We believe that this narrower definition is appropriate to avoid too many clinical practice variations in COVID-19 diagnosis, especially in the early years of the pandemic.

---

## [Decision Letter · Decision Letter 5]

17 Apr 2024

PONE-D-22-31470R5Hospital services utilisation and cost before and after COVID-19 hospital treatment: evidence from IndonesiaPLOS ONE

Dear Dr. Hafidz,

Thank you for submitting your manuscript to PLOS ONE. After careful consideration, we feel that it has merit but does not fully meet PLOS ONE’s publication criteria as it currently stands. Therefore, we invite you to submit a revised version of the manuscript that addresses the points raised during the review process.

Dear author(s),

The seventh reviewer asked to amend all the main comments he requested in the fourth review round, and you did not amend what was required.

Either you are not convinced by those comments and are against them, or you did not fully understand what the reviewer wanted from you in order to amend it as necessary.

Therefore, I ask you to review the comments of the seventh reviewer, amend and include all the comments and concerns raised by the reviewer, and submit the manuscript for the sixth time for review.

Please carefully review all comments and do not ignore any of the comments already included in the previous peer review report.

Please see below the reviewer's 7 comments:

I'm not impressed by the authors' response to my comments and suggestions. They either do not want to understand my two main comments—and if so, they should say so or state that they disagree with me (which could be fine, of course)—or their understanding of the method they are applying (DID) is very shallow.

In particular, in responding to my first main comment on implications for interpretations of using 2021 cases as a comparison group, they keep repeating that they assume that the comparison group had not had COVID before. While I agree that this is an important assumption, my point was another one: The DID they run looks at how changes from before to after COVID differ for those having it in 2020 vs. 2021; see my prior report to the authors for elaboration.

I'm also not impressed with the shallow response to my second main comment.

Overall, I maintain my comment in my previous report that the interpretations they make do not sufficiently reflect the research question actually addressed by the method they in fact applied. As noted in my previous response to you, I do not think there is a need for new analyses, only that the interpretation and framing of the paper should align better with what the applied method (DID) is in fact addressing.

On the other hand, it seems to me that the authors have had to deal with 7 (?!) reviewers and a number of revisions, so I can understand if they are getting fed up with revising this paper. Thus, I see that making a decision on this paper can be a hard call for the editor.

We look forward to receiving your revised manuscript.

Kind Regards,

Asst. Prof. Dr. Nemer Badwan

Ph.D in Economics and Finance

Academic Editor and Reviewer

PLOS ONE

Journal Requirements:

Additional Editor Comments (if provided):

Reviewers' comments:

Reviewer's Responses to Questions

**Comments to the Author**

1. If the authors have adequately addressed your comments raised in a previous round of review and you feel that this manuscript is now acceptable for publication, you may indicate that here to bypass the “Comments to the Author” section, enter your conflict of interest statement in the “Confidential to Editor” section, and submit your "Accept" recommendation.

Reviewer #5: All comments have been addressed

Reviewer #7: (No Response)

2. Is the manuscript technically sound, and do the data support the conclusions?

Reviewer #5: Yes

Reviewer #7: Partly

3. Has the statistical analysis been performed appropriately and rigorously? 

Reviewer #5: Yes

Reviewer #7: I Don't Know

4. Have the authors made all data underlying the findings in their manuscript fully available?

Reviewer #5: Yes

Reviewer #7: No

5. Is the manuscript presented in an intelligible fashion and written in standard English?

Reviewer #5: Yes

Reviewer #7: Yes

6. Review Comments to the Author

Reviewer #5: (No Response)

Reviewer #7: (No Response)

7. PLOS authors have the option to publish the peer review history of their article (what does this mean?). If published, this will include your full peer review and any attached files.

Reviewer #5: No

Reviewer #7: No

---

## [Author Response · Author response to Decision Letter 5]

16 May 2024

Response to Reviewer #7

Thanks for your reply to our responses in the previous review report. We would like to elaborate more on them as the reviewer appears to see our previous replies to be insufficient. 

First, what we understand is that your first main concern is related to the application and interpretation of our DiD event-study approach. You concluded that we estimated the “effects on utilisation of COVID in 2020 vs 2021”. We would like to clarify that this is not what we meant.

As illustrated in Figure 1, we conducted four separate DiD event-study estimations, for cohort of individuals who were treated with COVID in May, June, July, and August 2020. We would like to explain it as clearly as possible here using the example of the event-study which examining the impact of COVID-19 for those hospitalised with SARS-CoV-2 in May 2020 (the top left panel of Figure 3). 

As stated in our statistical analysis, “we used the fifth month leading to the hospitalisation as the reference period.” 

New text: “For example, May 2020 group, this reference period was December 2019 and labelled as t-5. The t0 was May 2020, the t+1 to t+6 was June to November 2020, the t-4 to t-1 was January 2020 to April 2020. For each subsequent month, the reference and comparative periods shifted accordingly”.

Overall, it appears that we have disagreements around the implementation and interpretation of the DiD approach in our analysis. First, the interpretation of estimates before and after COVID-19 is from the perspective of those who were treated with COVID-19 between May and August 2020, in comparison to those who "will" eventually get COVID-19 in February 2021. In this case, we believe that our explanation is clear enough to highlight that the comparison group has not contracted COVID-19 in the analytical period. 

New Text in Study Population Section: " The comparison group comprised individuals treated for COVID-19 in hospitals in February 2021 (not hospitalised in 2020), under the assumption they had not contracted the virus in 2020. We argue this is a suitable comparator for our analysis as these groups are similar (i.e. attitude towards risk, preference, etc.). As we observed their health care utilisation in the entirety of our estimation sample from August 2019 to November 2020, the event study approach allows us to compare the outcomes between treatment and comparison groups over time. "

As for your second main comment, we apologise if we did not respond to it as detailed as you expected, especially the last part of your second comment ("It would also be important to inform the reader of the survival in the two groups (i.e., by providing death rates by 3-6 months or even the impact on death using the same regressions). Was survival measured for the same length of time in the comparison group?"). We would like to answer it as the following:

We agree that survival is an important factor that determines healthcare costs and utilisations. We have answered this concern by stating that our focus is on the survivors. Again, from the perspective of the COVID-19 group, within the analytical periods (August 2019 to November 2020), all individuals in the comparison group had not contracted COVID-19 yet. Therefore, no survival rate associated with COVID-19 can be inferred from the comparison group. To provide some context in terms of the recorded mortality of the COVID-19 group, we have added the following sentence to our Study Population subsection.

“This decision was made to focus on COVID-19 survivors, hence avoiding the potential issue of reduced healthcare use by the COVID-19 group due to their higher post-hospital treatment mortality. For context, the recorded 6-month mortality rate of those treated with COVID-19 between May and August 2020 was 12.1%”

---

## [Decision Letter · Decision Letter 6]

6 Jun 2024

Hospital services utilisation and cost before and after COVID-19 hospital treatment: evidence from Indonesia

PONE-D-22-31470R6

Dear Dr. Firdaus Hafidz,

We’re pleased to inform you that your manuscript has been judged scientifically suitable for publication and will be formally accepted for publication once it meets all outstanding technical requirements.

Kind Regards,

Asst. Prof. Dr. Nemer Badwan 

Ph.D in Economics and Finance

Assistant Professor of Economics and Finance 

Academic Editor and Reviewer 

PLOS ONE

Additional Editor Comments (optional):

Reviewers' comments:

Reviewer's Responses to Questions

**Comments to the Author**

1. If the authors have adequately addressed your comments raised in a previous round of review and you feel that this manuscript is now acceptable for publication, you may indicate that here to bypass the “Comments to the Author” section, enter your conflict of interest statement in the “Confidential to Editor” section, and submit your "Accept" recommendation.

Reviewer #5: All comments have been addressed

Reviewer #8: All comments have been addressed

2. Is the manuscript technically sound, and do the data support the conclusions?

Reviewer #5: Yes

Reviewer #8: Yes

3. Has the statistical analysis been performed appropriately and rigorously? 

Reviewer #5: Yes

Reviewer #8: Yes

4. Have the authors made all data underlying the findings in their manuscript fully available?

Reviewer #5: Yes

Reviewer #8: Yes

5. Is the manuscript presented in an intelligible fashion and written in standard English?

Reviewer #5: Yes

Reviewer #8: Yes

6. Review Comments to the Author

Reviewer #5: I have reviewed this manuscript before, and it has improved during the revisions. I have no further comments.

Reviewer #8: Dear authors,

thank you for this opportunity to read "Hospital services utilisation and cost before and after COVID-19 hospital treatment: evidence from Indonesia". It was very clear and well written.

My only concern, and it is minor and should not delay publication, is that in future articles, please be clear about the average outpatient visit costs or per capita expenditure, by island, age, and gender, should be if people were in good health. Reviewing your supplemental tables assisted in my understanding of your paper. Reporting that information will help future readers understand the magnitude and burden of increased utilization on the Indonesian health system.

Overall, a good paper. I look forward to seeing it in print.

7. PLOS authors have the option to publish the peer review history of their article (what does this mean?). If published, this will include your full peer review and any attached files.

Reviewer #5: No

Reviewer #8: No

---

## [Editor Report · Acceptance letter]

25 Jun 2024

PONE-D-22-31470R6 

PLOS ONE

Dear Dr. Hafidz, 

I'm pleased to inform you that your manuscript has been deemed suitable for publication in PLOS ONE. Congratulations! Your manuscript is now being handed over to our production team.

Kind regards, 

on behalf of

Asst. Prof. Dr. Nemer Badwan 

Academic Editor

PLOS ONE